# Expression of ALS-PFN1 impairs vesicular degradation in iPSC-derived microglia

Salome Funes[1,2,15], Jonathan Jung[1,3,15], Del Hayden Gadd[1], Michelle Mosqueda[4,5], Jianjun Zhong[1,6], Shankaracharya[1], Matthew Unger[5,7], Karly Stallworth[1], Debra Cameron[1], Melissa S. Rotunno[1], Pepper Dawes[8,9], Megan Fowler-Magaw[1,3], Pamela J. Keagle[1], Justin A. McDonough[10], Sivakumar Boopathy[1,5], Miguel Sena-Esteves[1,11], Jeffrey A. Nickerson[12], Cathleen Lutz[13], William C. Skarnes[10], Elaine T. Lim[3,8,9], Dorothy P. Schafer[3,14], Francesca Massi[4,5], John E. Landers[1,3] & Daryl A. Bosco[1,2,3,4,5] ✉

Microglia play a pivotal role in neurodegenerative disease pathogenesis, but the mechanisms underlying microglia dysfunction and toxicity remain to be elucidated. To investigate the effect of neurodegenerative disease-linked genes on the intrinsic properties of microglia, we studied microglia-like cells derived from human induced pluripotent stem cells (iPSCs), termed iMGs, harboring mutations in profilin-1 (PFN1) that are causative for amyotrophic lateral sclerosis (ALS). ALS-PFN1 iMGs exhibited evidence of lipid dysmetabolism, autophagy dysregulation and deficient phagocytosis, a canonical microglia function. Mutant PFN1 also displayed enhanced binding affinity for PI3P, a critical signaling molecule involved in autophagic and endocytic processing. Our cumulative data implicate a gain-of-toxic function for mutant PFN1 within the autophagic and endo-lysosomal pathways, as administration of rapamycin rescued phagocytic dysfunction in ALS-PFN1 iMGs. These outcomes demonstrate the utility of iMGs for neurodegenerative disease research and implicate microglial vesicular degradation pathways in the pathogenesis of these disorders.

Microglia are resident macrophages and primary phagocytes of the central nervous system (CNS). The phagocytic properties are critical for microglia function, including synapse pruning, circuit remodeling, and clearing debris[1]. While microglia serve to sculpt and protect the CNS microenvironment, microglia dysregulation is pathologically associated with multiple neurodegenerative diseases, including Alzheimer's disease, Parkinson's disease, amyotrophic lateral sclerosis (ALS), and frontotemporal dementia (FTD)[2]. Transcriptomic analyses indicate that microglia adopt an altered state in the context of neurodegeneration[3]. Further, the functional properties of microglia change in a neurodegenerative disease context, wherein microglia can exhibit increased production of inflammatory mediators and altered phagocytic behaviors that are, in turn, neurotoxic[2]. However, the

extent to which microglial dysfunction occurs as a response to a diseased CNS environment versus from cell-autonomous factors is not fully understood.

Rodent models have been integral for elucidating the physiological and pathological functions of microglia. However, inter-species variations often limit the biological relevance of microglia-specific findings for human disease[1]. Several protocols have recently emerged for the differentiation of human induced pluripotent stem cells (iPSCs) into microglia-like cells[4–6], which represent a valuable cellular model to study microglia processes related to human development and neurodegenerative disease. Human microglia-like cells also represent an ideal system for investigating disease-associated phenotypes[7], including whether disease-linked genes affect the intrinsic properties

of microglia[8]. For the current study, we used our optimized protocol that mimics microglia Myb-independent ontogeny for the generation of human iPSC-derived microglia-like cells, which we refer to as iMGs, in sufficient yield for downstream gene expression and functional analyses[9,10]. Notably, iMGs produced from our protocol express microglia-enriched markers, exhibit a transcriptome that resembles primary human microglia, respond to immune stimulation, and exhibit phagocytosis.

Herein, we used iMGs to investigate the role of profilin 1 (PFN1) in ALS[11], a fatal neurodegenerative disorder. PFN1 plays an essential role in modulating actin dynamics through interactions with actin, proteins enriched with poly-L-proline motifs, and phosphoinositide (PIP) lipids[12]. Autosomal dominant mutations in PFN1 are highly penetrant and account for 1-2% of inheritable ALS[11,13]. Further, cytoskeletal dysfunction is a major pathophysiological feature of both familial and sporadic ALS[11]. ALS-PFN1 predominantly presents with limb onset and results in a similar disease course as other forms of ALS, however the mechanism underlying PFN1-mediated ALS has not been elucidated. Although ALS is primarily classified by upper and lower motor neuron degeneration, microglial pathology is commonly observed in ALS patients and animal models[14]. Intriguingly, *PFN1* transcripts are more abundant in microglia than neurons[15], raising the possibility that PFN1 plays important physiological roles in microglia. In support of this notion, PFN1 becomes upregulated in reactive microglia following brain injury[16] and PFN1 knockdown attenuates the microglia proinflammatory state in models of ischemia[17]. PFN1 was also identified as a modifier of phagocytosis through a CRISPR screen in human iPSC-derived microglia-like cells[18]. Given these observations, we sought to determine whether ALS-linked mutations in PFN1 alter the intrinsic properties of iMGs.

Our proteomic and transcriptomic analyses of ALS-PFN1 versus WT iMGs revealed differentially expressed proteins and genes, respectively, related to lipid metabolism, autophagy, and phagocytosis. Autophagy and phagocytosis represent distinct yet related vesicular degradation pathways that coordinate packaging of intracellular (autophagy) and extracellular (phagocytosis) substrates into vesicles, which ultimately fuse with the lysosome for degradation[19]. ALS-PFN1 iMGs also exhibited evidence of autophagy dysregulation with concomitant accumulation of lipid droplets, which should normally be cleared through the autophagy pathway. While ALS-PFN1 iMGs were able to engulf synaptosomes and other substrates, mutant PFN1 iMGs were deficient in processing phagocytosed material through the endo-lysosomal pathway. Herein, we used nuclear magnetic resonance (NMR) to examine the binding interaction between PFN1 and PI3P, a phospholipid that plays critical roles in autophagy and phagocytosis[20,21]. Our cumulative data implicate a gain-of-toxic mutant PFN1 function in the context of microglial vesicular degradation, which could be pharmacologically ameliorated with rapamycin. While much attention has focused in the endo-lysosomal and autophagy pathways in neurons, disruption of these pathways within microglia may also contribute to neurodegenerative disease pathogenesis[22].

## Results

### Generation and characterization of human ALS-PFN iPSC-derived microglia-like cells (iMGs)

ALS-linked C71G[+/-] and M114T[+/-] mutations were introduced into the *PFN1* locus of the human iPSC KOLF2.1J line using CRISPR/Cas9 gene-editing[13,23]. In addition, the homozygous M114T[+/+] variant was created as an experimental line to investigate mutant-gene dosage on potential phenotypes (Supplementary Fig. 1). Mutant iPSCs and their respective WT isogenic control lines were differentiated into iMGs using previously published protocols with some modifications (Fig. 1a)[6,10]. Briefly, iPSCs were differentiated into embryoid bodies (EBs). After 21–28 days, EBs produced primitive macrophage precursors (PMPs) that were terminally differentiated into iMGs for 10 to 12 days[5,6]. As in

other reports of iPSC-derived microglia, iMGs exhibited a microglia-like morphology with some ramifications and a relatively small cytoplasmic area[4,5]. Immunofluorescence analyses determined that >90% of iMGs expressed the microglia-signature proteins purinergic receptor P2RY12 and transmembrane protein 119 (TMEM119), as well as the myeloid marker protein ionized calcium-binding adapter molecule 1 (IBA1) (Fig. 1b and Supplementary Fig. 2a). Expression of "homeostatic" microglia-signature genes including G protein-coupled receptor 34 (*GPR34*), protein S (*PROS1*), *P2RY12* and MER proto-oncogene tyrosine kinase (*MERTK*)[24] were significantly increased in iMGs compared to PMPs and iPSCs as determined by qPCR (Fig. 1c). Expression of microglia-signature genes was further substantiated by RNA-sequencing (RNAseq) analysis of mutant PFN1 and WT iMGs derived from five different iPSC lines, including PFN1 C71G[+/-], M114T[+/-] and two WT lines (Supplementary Fig. 2b, c). Additionally, expression of the myeloid transcription factor PU.1 gene (*SPI-1*) was higher in PMPs relative to iMGs, but negligible in iPSCs. Expression of the pluripotency marker SRY-Box transcription factor 2 (*SOX2*) was significantly reduced in iMGs (Fig. 1c). None of the aforementioned genes were expressed differentially with statistical significance between WT and ALS-PFN1 iMGs (Data S1).

Outcomes of the RNAseq analysis were compared to publicly available RNASeq data sets from other studies of iPSC-derived microglia[4,5]. Principal component analysis indicated that the transcriptome of our iMGs most closely resemble primary human microglia and were most divergent from iPSCs and monocytes (Fig. 1d). We also evaluated genes that reportedly change as a function of microglia developmental stage[25] (Supplementary Fig. 2b). Read counts for genes associated with adult microglia, including V-Maf musculoaponeurotic fibrosarcoma oncogene homolog B (*MAFB*), monocyte differentiation antigen CD14 (*CD14*), and the colony-stimulating factor-1 receptor (*CSFR1*) were higher than for genes associated with fetal microglia, such as minichromosome maintenance complex component 5 (*MCM5*). Only one gene associated with yolk-sac microglia progenitors (coagulation factor XIII A chain, *F13A1*) was detected in iMGs and, as expected, with low abundance[25]. Notably, abundant levels of *APOE* and *SPP1*, genes that are highly expressed in aged microglia, were detected in both mutant PFN1 and WT iMGs[26].

The reactivity of iMGs to lipopolysaccharide (LPS) was then assessed by quantifying cytokine release. At baseline, WT, PFN1 C71G[+/-] (Fig. 1e), M114T[+/-] and M114T[+/+] (Supplementary Fig. 3) iMGs secreted low levels of the pro-inflammatory cytokines interleukin (IL)−6, tumor necrosis factor-alpha (TNF-α) and regulated on activation, normal T cell expressed and secreted (RANTES or CCL5) as well as the anti-inflammatory cytokine IL-10. The levels of IL-6, CCL5, and IL-10 increased significantly 24h-post LPS treatment, whereas TNF-α levels were significantly elevated after 6 h of LPS stimulation. The cytokine levels were generally similar between mutant and WT PFN1 cells, except IL-6 was moderately elevated in M114T lines after 6 h of LPS stimulation and IL-10 reduced in M114T[+/+] at baseline (Fig. 1e, Supplementary Fig. 3 and Data S1). Collectively, these data show that iMGs from both genotypes are responsive to immune stimulation.

### Factors involved in lipid metabolism and vesicular degradation are altered in ALS-PFN1 iMGs

Next, we investigated proteins and genes that were differentially expressed between ALS-PFN1 and WT iMGs. Quantitative proteomics with 6-plex tandem mass tag (TMT) labeling of whole cell lysates was used to assess differential protein expression between PFN1 C71G[+/-] and WT iMGs by mass spectrometry. A total of 1813 proteins in 1660 clusters were detected in the TMT analysis, with 56 proteins differentially expressed between PFN1 C71G[+/-] and WT iMGs (Fig. 2a and Data S2). Of those 56 proteins, 30 proteins were upregulated in PFN1 C71G[+/-] iMGs including fatty acid-binding protein (FABP) 4 and FABP5. PFN1 was the most significantly downregulated protein. Consistent with

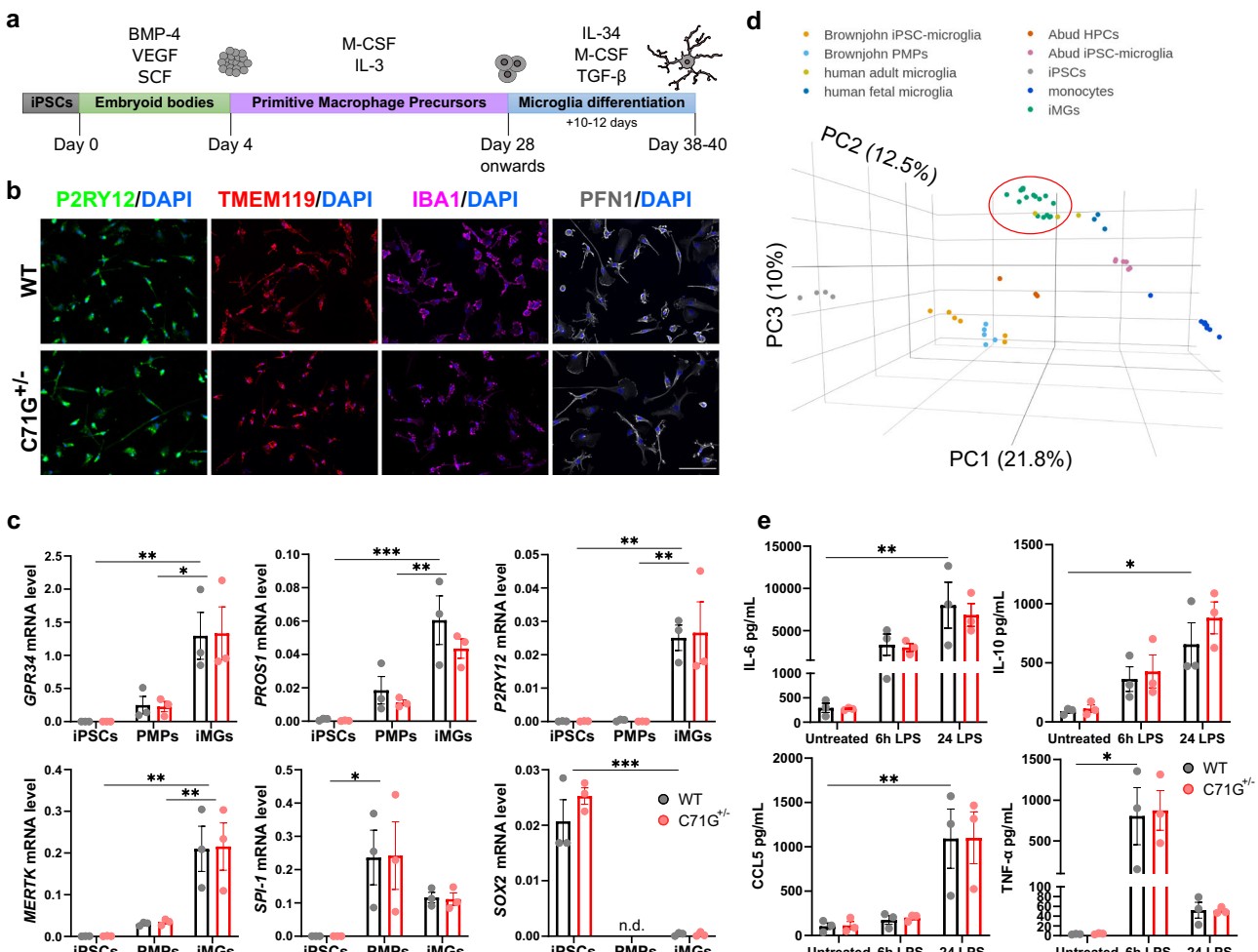

**Fig. 1 | Generation and characterization of WT and ALS-PFN1 iPSC-derived microglia-like cells (iMGs). a** Schematic representation of iMG differentiation protocol. Bone morphogenetic protein 4 (BMP-4), vascular endothelial growth factor (VEGF), and stem cell factor (SCF) are used for embryoid body differentiation. Macrophage colony-stimulating factor (M-CSF) and interleukin (IL)−3 are included during PMP generation. IL-34, M-CSF, and transforming growth factor beta (TGF-β) are added during terminal iMG differentiation. Created with BioRender.com. **b, c** WT and C71G$^{+/-}$ iPSCs differentiated into iMGs. **b** Representative immunofluorescence images of P2RY12, TMEM119, and PFN1 for $n = 3$ independent differentiations. Scale bar: 100 μm. **c** Comparison of gene expression levels between iPSCs, PMPs, and iMGs of myeloid and microglia-enriched genes including *GPR34* (*$P = 0.0173$, ** $P = 0.0044$), *PROS1* (**$P = 0.0040$, ***$P = 0.0002$), *P2RY12* (**$P = 0.0030$ for iPSC vs iMG, **$P = 0.0033$ for PMP vs iMG), *MERTK* (**$P = 0.0016$ for iPSC vs iMG, **$P = 0.0049$ for PMP vs iMG), and *SPI-1* (*$P = 0.0240$) and the pluripotency marker *SOX2* (***$P = 0.0003$) **d** Three-dimensional principal component analysis (PCA) of iMGs from this study (green; WT $n = 7$, C71G$^{+/-}$ $n = 4$, M114T$^{+/-}$

$n = 3$, M114T$^{+/+}$ $n = 3$, where $n =$ an independent differentiation), primary human adult (yellow) and fetal microglia (navy blue) from Abud et al.[4] iPSC-derived microglia-like cells (pink) and the intermediate hematopoietic progenitors cells (HPCs, dark orange) from Abud et al.[4] iPSC-derived microglia-like cells (light orange) and the intermediate progenitors PMPs (light blue) from Brownjohn et al.[5] and monocytes (dark blue) and iPSCs (gray) from Abud et al.[4]. Variance of each principal component (PC) is indicated in parenthesis along the axes. **e** WT and C71G$^{+/-}$ iMGs secrete elevated levels of IL-6 (**$P = 0.0050$), IL-10 (*$P = 0.0156$), CCL5 (**$P = 0.0077$) and TNF-α (*$P = 0.0208$) after 6 h or 24 h of 100 ng/mL LPS stimulation compared to untreated cells. Statistics were determined by two-way ANOVA and Šídák's multiple comparisons test. No WT vs C71G$^{+/-}$ comparisons were statistically significant. *P*-values are listed only for WT cells for simplicity; all other statistical comparisons are defined in Data S1. Mean ± SEM for $n = 3$ independent differentiations are shown for all bar graphs, with each data point representing an individual differentiation. Source data are provided as a Source Data file.

previous findings in PFN1 C71G$^{+/-}$ and M114T$^{+/-}$ patient-derived lymphoblast cells[27,28], all mutant PFN1 iMGs expressed lower PFN1 levels compared to control cells by Western blot analysis, with ~60% reduction in PFN1 C71G$^{+/-}$ iMGs (Fig. 2b, c) and ~30% reduction in PFN1 M114T$^{+/-}$ iMGs (Fig. 2b, d). Strikingly, PFN1 levels were reduced by 80% in M114T$^{+/+}$ iMGs compared to WT iMGs (Fig. 2b, d). These results are consistent with the destabilizing effect of ALS-linked mutations on PFN1 structure, where the C71G variant is more misfolded than the M114T variant and thus is more robustly degraded by the proteasome[27,29].

Enrichr and Metascape were used to perform functional enrichment analyses of the 56 differentially expressed proteins. There was overlap between both approaches with common terms related to

lipids, ferroptosis, and phagosome. The Bioplanet pathway analysis from Enrichr highlighted lipid metabolism and transport (Fig. 2e and Data S3). Enrichr also identified "ferroptosis", a form of iron-dependent cell death caused by accumulation of lipid peroxides[30], as the most significantly enriched term from the KEGG pathway library (Supplementary Fig. 4a and Data S4). Additional lipid-related terms, including "lipid droplets" and "fatty acid transporters" were among the top enriched terms from GO cellular components and Metascape analyses, respectively (Supplementary Fig. 4b, c and Data S4).

Considering these lipid-related terms, we probed for evidence of lipid accumulation in the form of lipid droplets. Lipid droplets contain a core of neutral lipids surrounded by a phospholipid monolayer. While lipid droplets represent lipid storage sites under physiological

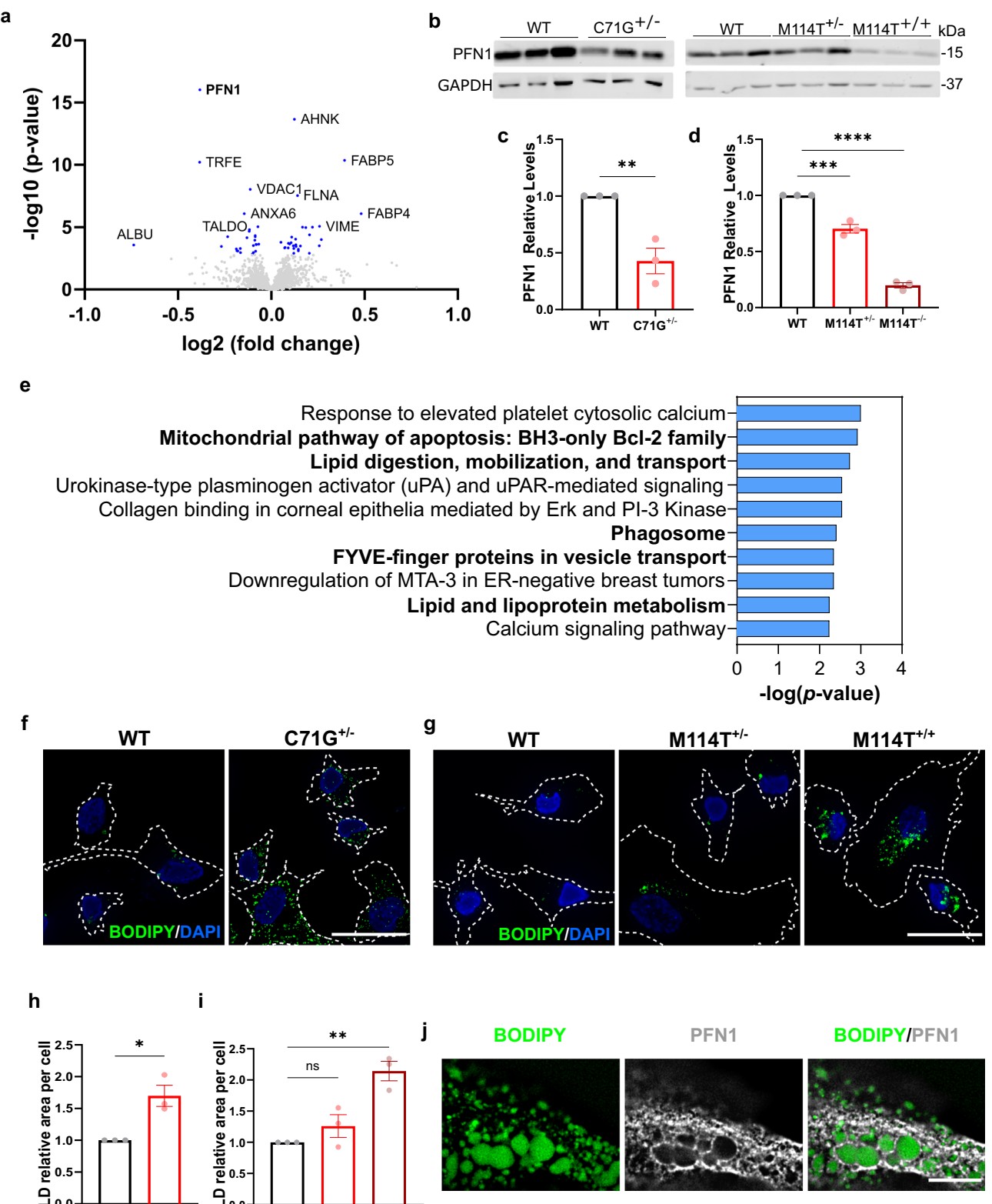

conditions, aberrant accumulation of lipid droplets correlates with aging and disease[31,32]. Mutant PFN1 and WT iMGs were stained with BODIPY (Fig. 2f, g), a fluorescent dye that binds lipid droplets, and the BODIPY signal was quantified by fluorescence microscopy. Mutant PFN1 iMGs presented with more BODIPY signals compared to WT iMGs, especially in PFN1 C71G$^{+/-}$ (Fig. 2h) and M114T$^{+/+}$ (Fig. 2i) iMGs. A low percentage (<5%) of both ALS-PFN1 and WT iMGs exhibited a dense area of lipid droplets, which appeared to fill most of the cell. Upon co-

staining for BODIPY and anti-PFN1, PFN1 signal was clearly observed surrounding lipid droplets in these cells (Fig. 2j). While PFN1 may also be associated with lipid droplets in cells presenting with fewer lipid droplets, we note that detection of PFN1 surrounding lipids droplets was most obvious in iMGs with high lipid droplet content. These observations raise the possibility that PFN1 associates with lipid droplets, potentially through direct binding of PFN1 with the phospholipid outer shell.

**Fig. 2 | Quantitative proteomics analysis of ALS-PFN1 iMGs. a** Volcano plot of proteins differentially expressed between C71G$^{+/-}$ and WT iMGs identified by TMT proteomics. Log2 fold change in expression and significance (−log10 $p$-value) are displayed on the x- and y-axes, respectively. Differentially expressed proteins with a $P$-value < 0.00160 (blue dots) are considered significant after T-test followed by Benjamini-Hochberg correction (see Data S2). **b** Western blot analysis of PFN1 levels in C71G$^{+/-}$, M114T$^{+/-}$, M114T$^{+/+}$ iMGs compared to control WT iMGs. **c**, **d** Quantification of (**b**). For each independent differentiation, PFN1 protein levels were normalized first to GAPDH (loading control) and then to the respective WT control from the same differentiation. Statistics were determined using unpaired two-tailed t-test for **c** (**$P$ = 0.0073, t = 5.035, df = 4) and ordinary one-way ANOVA with Dunnett's multiple comparisons test for **d** (***$P$ = 0.0003, $q$ = 8.232, df = 6, and ****$P$ < 0.0001, $q$ = 22.20, df = 6). **e** Bar graph of enriched terms across differentially expressed proteins generated by Enrichr using the Bioplanet 2019 library (see Data S3). The significance of the term is defined on the x-axis. Terms of interest are emboldened. $P$-values were computed using a two-

sided Fischer's exact test. **f–j** Lipid droplets accumulate in mutant PFN1 iMGs as determined by BODIPY immunofluorescence analysis. **f, g** Representative immunofluorescence images of BODIPY staining in **f** PFN1 WT and C71G$^{+/-}$ iMGs and **g** PFN1 WT, M114T$^{+/-}$ and M114T$^{+/+}$ iMGs. Cell boundaries are depicted with white dashed lines. Scale bar = 25 μm. **h, i** Quantification of the area of BODIPY fluorescence signal representing lipid droplets (LD) was normalized to WT iMGs within each independent differentiation for **f** (unpaired two-tailed t-test, *$P$ = 0.0137, t = 4.201, df = 4) and **g** (ordinary one-way ANOVA and Dunnett's multiple comparisons test, ns $P$ = 0.3647, $q$ = 1.338, df = 6 for WT vs M114T$^{+/-}$ and **$P$ = 0.0020, $q$ = 5.840, df = 6 for WT vs M114T$^{+/+}$). **j** Representative immuno-fluorescence images (one Z-plane acquired by focusing on prominent lipid droplets) showing PFN1 (gray) surrounding lipid droplets (BODIPY in green) in C71G$^{+/-}$ iMGs from $n$ = 3 independent differentiations. Scale bar = 10 μm. All data include $n$ = 3 independent differentiations. Bar graphs show mean ± SEM with individual data points representing independent differentiations. Source data are provided as a Source Data file.

Next, PFN1 knockdown studies were pursued to determine whether lipid droplet accumulation is a consequence of PFN1 loss-of-function. As lentiviral transduction was toxic to iMGs, these studies were carried out in the human microglia immortalized HMC3 line using an antibody against perilipin 2, a prominent lipid droplet-associated protein[31]. Lipid droplet content was not elevated or significantly different upon PFN1 knockdown compared to control cells (Supplementary Fig. 5), implicating a gain-of-toxic function for mutant PFN1 with respect to lipid dysmetabolism in iMGs. Taken together, the trend in M114T$^{+/-}$ < C71G$^{+/-}$ < M114T$^{+/+}$ iMGs for lipid droplet content likely reflects that C71G$^{+/-}$ is more misfolded than M114T$^{+/-}$ and that the relative levels of misfolded protein are highest in M114T$^{+/+}$ iMGs.

A differential gene expression analysis was also performed with RNASeq data from ALS-PFN1 iMGs, which included both PFN1 C71G$^{+/-}$ and M114T$^{+/-}$ lines, compared to the isogenic WT lines. Although only thirteen genes were differentially expressed with statistical significance, thus precluding a pathway analysis, *TBC1D15* emerged as a highly significant, upregulated gene in ALS-PFN1 iMGs (Fig. 3a and Data S5). TBC1D15 is a GTPase-activating protein for RAB7 and is involved in autophagosome biogenesis during mitophagy, a form of selective autophagy that degrades mitochondria[33]. Elevated TBC1D15 was confirmed at the mRNA and protein level by qPCR (Fig. 3b, c) and Western blot (Fig. 3d, e) analyses, respectively. No difference in Rab7 levels were detected between WT and mutant PFN1 iMGs (Supplementary Fig. 6), although this does not preclude the possibility that Rab7 activity could be altered in these cells. In contrast to ALS-PFN1 iMGs, knockdown of PFN1 in HMC3 cells resulted in significantly reduced levels of TBC1D15 (Supplementary Fig. 7), implicating a physiological relationship between PFN1 WT and TBC1D15. RASD family member 2 (*RASD2*)[34], which is also involved in autophagy, was elevated in mutant PFN1 iMGs in the RNASeq (Fig. 3a) and validated by Western blot (Supplementary Fig. 6).

Given that both TBC1D15 and RASD2 play a role in mitophagy[33,34], we probed for colocalization of TBC1D15 with TOMM20, an outer mitochondrial membrane marker, by immunofluorescence (Fig. 3f). TBC1D15 and TOMM20 showed an increase in colocalization in PFN1 C71G$^{+/-}$ iMGs compared to WT iMGs as indicated by the Pearson's correlation (Fig. 3g). As the Pearson's correlation revealed enhanced colocalization between PFN1 and TBC1D15 in mutant iMGs, but not between PFN1 and TOMM20 (Supplementary Fig. 8a, b, c), it is unlikely that PFN1 directly increases TBC1D15/TOMM20 colocalization in mutant iMGs. Total levels of mitochondria were not significantly different between mutant and control iMGs (Supplementary Fig. 8d, e). Therefore, there appears to be an increase in signals marking mitochondria for autophagosomal degradation in PFN1 C71G$^{+/-}$ iMGs, but without an overall change in mitochondrial content.

## ALS-PFN1 microglia exhibit deficits in vesicular degradation of phagocytosed material in vitro and in vivo

A critical function of microglia is phagocytosis, a process that was also highlighted by our proteomics analysis (Fig. 2e). We examined phagocytosis in iMGs using live-cell imaging with substrates that were labeled with pHrodo red (pHrodo), a pH-sensitive dye that emits red fluorescence exclusively in low pH cellular compartments. Low pH cellular compartments include late endosomes and phagolysosomes, the latter being generated upon fusion of phagosomes and lysosomes during phagocytosis[35]. Initially, we administered pHrodo-labeled mouse synaptosomes to PFN1 C71G$^{+/-}$ and WT iMGs, resulting in enhanced red fluorescent signals in both lines by 30 min that became more pronounced by 2 h (Fig. 4a). Quantification of a phagocytosis index for each line revealed a significant increase in pHrodo signal in PFN1 C71G$^{+/-}$ versus WT iMGs over the course of 12 h. Addition of cytochalasin D, an inhibitor of actin polymerization that prevents phagocytosis, significantly attenuated pHrodo fluorescence (Fig. 4b, c)[36]. Similar to PFN1 C71G$^{+/-}$, PFN1 M114T$^{+/-}$ and M114T$^{+/+}$ iMGs also produced enhanced pHrodo fluorescence relative to their WT counterparts in this assay. To verify that the pHrodo-labeled signal was originating from intracellular acidic compartments, we pre-treated iMGs with bafilomycin A (BafA), a vacuolar H$^+$-ATPase inhibitor that blocks acidification of the phagolysosome[37]. As expected, red fluorescent signal is undetected in all BafA-treated iMGs (Fig. 4d, e). Similar phagocytosis assays were conducted with other disease-relevant substrates including human synaptosomes isolated from iPSC-derived lower motor neurons and aggregated C71G PFN1 recombinant protein. Administration of both substrates resulted in a significantly higher phagocytosis index in PFN1 C71G$^{+/-}$ iMGs relative to WT iMGs (Supplementary Fig. 9).

Based on the phagocytosis assays with pHrodo-labeled material alone, it is unclear whether higher pHrodo signal in mutant iMGs stems from enhanced substrate uptake or inefficient substrate degradation. To distinguish between these possibilities, a synaptosome washout step was included in the phagocytosis assay, allowing for quantification of the initial uptake of synaptosomes as well as the amount of undegraded synaptic material after 48 h post-washout (Fig. 4f). Human synaptosomes were used here, as iMGs processed these more rapidly than murine synaptosomes. To track synaptosomes in the initial phase of the assay, iMGs were incubated with constitutively fluorescent synaptosomes for 15 min, after which unbound material was washed away (referred to as 0 h post-washout). Alexa fluor-labeling was used instead of pHrodo, as pHrodo cannot be used to monitor phagocytosed material before it enters acidic compartments. An engulfment index was calculated based on the area of synaptosomes in direct contact with and inside iMGs, as delimited by anti-IBA1 staining (Fig. 4g), and was found to be similar for PFN1 C71G$^{+/-}$ and WT iMGs (Fig. 4h). To substantiate and

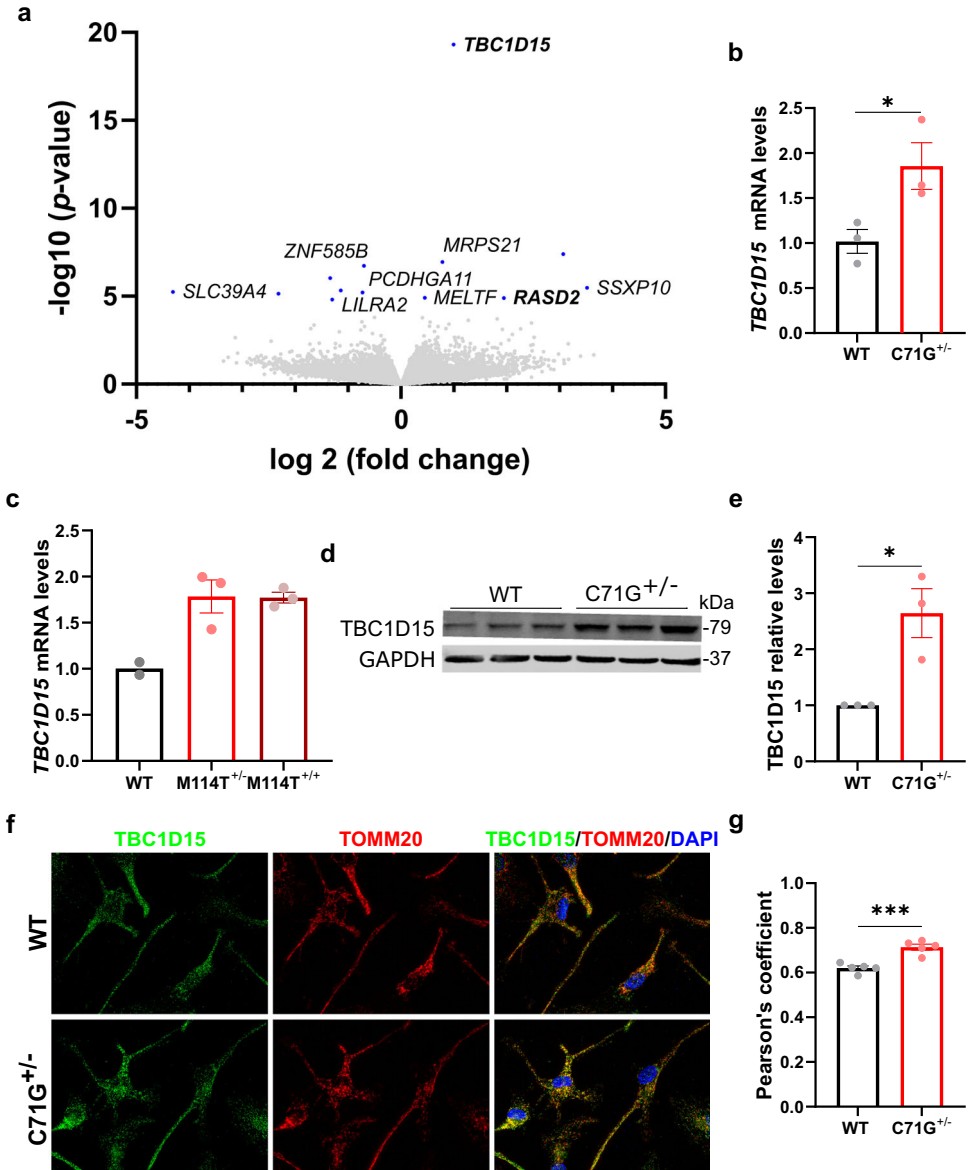

**Fig. 3 | TBC1D15 is upregulated in ALS-PFN1 iMGs and colocalizes with mitochondria. a** Volcano plot of differentially expressed genes between ALS-PFN1 iMGs ($n = 4$ for C71G[+/-] iMGs and $n = 3$ for M114T[+/-] iMGs) relative to WT iMGs ($n = 7$), where n refers to an independent differentiation; see Data S5. Differentially expressed genes (*P*-adjusted value < 0.05) are highlighted in blue. **b, c** Relative mRNA levels of *TBC1D15* normalized to the average of the respective WT iMGs. **b** *P* = 0.0452, $t = 2.875$, df = 4 for $n = 3$ independent differentiations; **c** WT $n = 2$, M114T[+/-] and M114T[+/+] $n = 3$ independent differentiations. **d** TBC1D15 protein expression determined by Western blot analysis. **e** Quantification of (**d**). TBC1D15 levels were normalized to GAPDH and then to the levels of the respective WT line from the same differentiation (*P* = 0.0197, $t = 3.763$, df = 4) for $n = 3$

independent differentiations. **f, g** Colocalization analysis of TBC1D15 and TOMM20 in WT and C71G[+/-] iMGs. **f** Representative immunofluorescence images of TBC1D15 (green), TOMM20 (red), and merged images including DAPI (blue) showing regions of overlayed TBC1D15:TOMM20 signal in yellow. Scale bar = 25 μm. **g** Pearson's correlation coefficient of TBC1D15 and TOMM20 signal (***P* = 0.0004, $t = 5.714$, df = 8) for $n = 5$ independent differentiations. All bar graphs show mean ± SEM, where each data point represents an independent differentiation. Statistics were determined using unpaired two-tailed *t*-test for WT vs C71G[+/-] iMGs comparisons or ordinary one-way ANOVA with Dunnett's multiple comparisons test for WT vs M114T[+/-] and M114T[+/+] comparisons. Source data are provided as a Source Data file.

extend these results, we also measured synaptosome uptake by flow cytometry. Consistent with the IF results, initial synaptosome uptake was similar across PFN1 C71G[+/-], M114T[+/-], M114T[+/+] and WT iMGs. This assay also showed that BafA attenuates but does not prevent synaptosome uptake (Fig. 4i and Supplementary Fig. 10). Therefore, lack of pHrodo fluorescence in the presence of BafA (Fig. 4d, e) can be attributed to inhibition of lysosome acidification and not due to inhibition of synaptosome uptake.

Given that actin polymerization is required during the early stages of phagocytosis and that ALS-linked PFN1 variants can modify actin polymerization[27,38], we also measured F-actin levels with phalloidin in

fixed-cells at baseline (i.e., in the absence of synaptosomes) and at 0 h post-washout. A redistribution of F-actin signal was observed in both PFN1 C71G[+/-] and WT iMGs upon addition of synaptosomes, but there were no discernable differences in F-actin signals between genotypes (Supplementary Fig. 11). Therefore, it is unlikely that deficits in synaptosome uptake or actin polymerization underlie the differences in phagocytosis between mutant PFN1 and WT iMGs.

At 48 h post-washout (Fig. 4f), the volume of pHrodo-labeled synaptosomes was quantified within compartments that stained for cluster of differentiation 68 (CD68), a macrophage/microglia-specific endo-lysosomal marker (Fig. 4j). PFN1 C71G[+/-] iMGs contained a

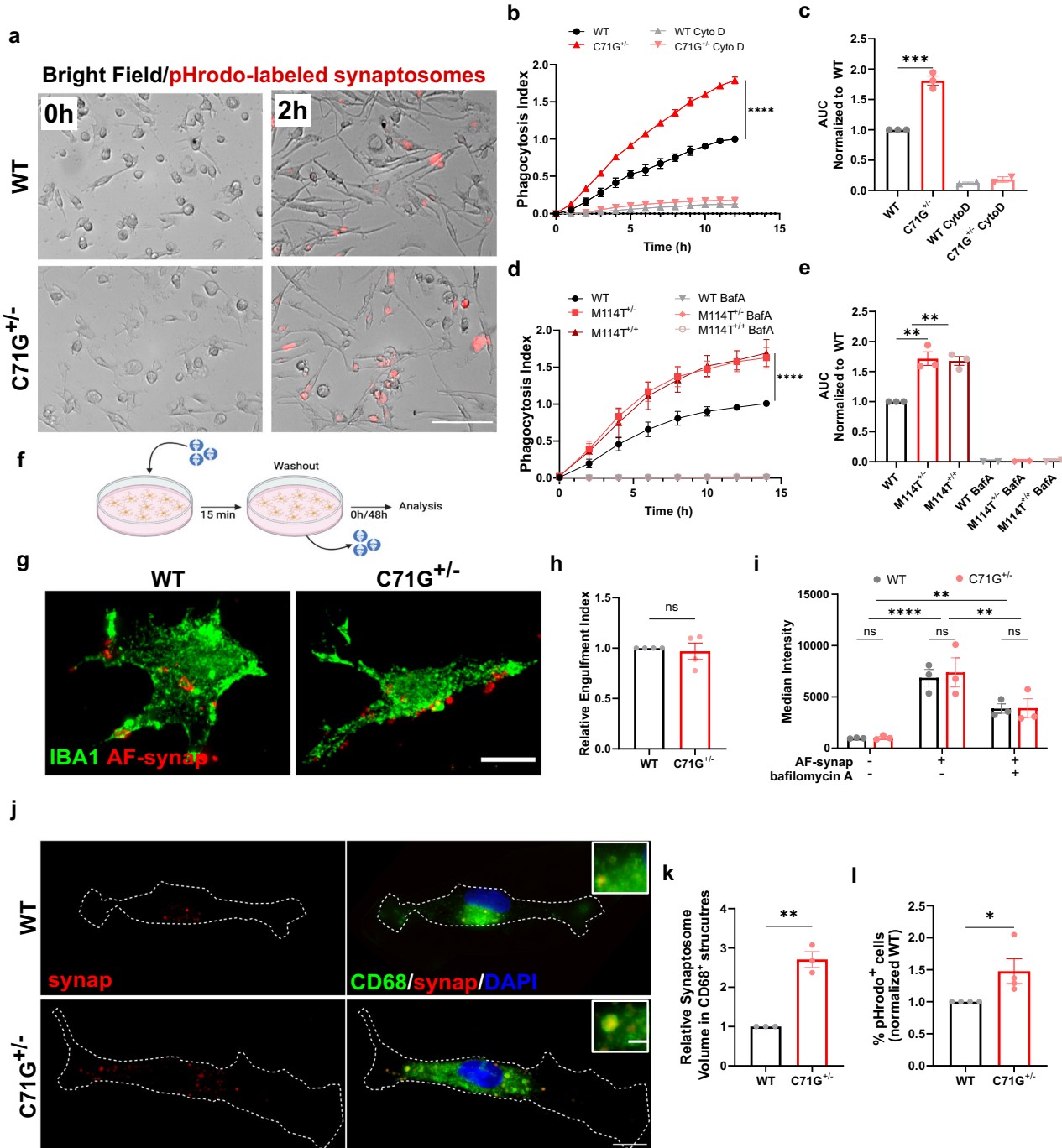

**Fig. 4 | ALS-PFN1 iMGs exhibit inefficient phagocytic degradation.**
**a**–**e** Phagocytosis of pHrodo-labeled mouse synaptosomes. **a** Representative images from the phagocytosis assay. Scale bar = 100 μm. **b** Quantification of the phagocytosis index (****$P < 0.0001$, two-way ANOVA with Sidak's multiple comparison test, ($F(1, 52) = 823$) for $n = 3$ independent differentiations). iMGs ($n = 2$) were pretreated with 10 μM Cytochalasin D (CytoD). **c** Area under the curve (AUC) determined from **b** (***$P = 0.0004$, unpaired two-tailed $t$-test $t = 10.60$, df = 4). **d** As in (**b**) with the indicated genotype for $n = 3$ independent differentiations (****$P < 0.0001$, $q = 7.434$, df = 48 for WT vs M114T$^{+/-}$ and ****$P < 0.0001$, $q = 7.236$, df = 48 for WT vs M114T$^{+/+}$, two-way ANOVA with Dunnett's multiple comparison test, F (2, 48) = 35.89). iMGs ($n = 2$) were pre-treated with 100 nM bafilomycin A (BafA). **e** AUC from (**d**) (**$P = 0.0011$, q = 6.536, df = 6 for WT vs M114T$^{+/-}$ and **$P = 0.0015$, q = 6.202, df = 6 for WT vs M114T$^{+/+}$). **f**–**k** Washout assay using human synaptosomes. **f** Assay schematic. Created with BioRender.com **g**, **h** Assay with Alexa Fluor-synaptosomes (AF-synap) at 0 h post-washout. **g** Representative immunofluorescence images of iMGs labeled with IBA1 (green) engulfing AF-synap (red). Scale bar: 10 μm. **h** Quantification of the engulfment index normalized to WT iMGs

(ns $P = 0.7185$, unpaired two-tailed $t$-test, $t = 0.3779$, df = 6) for $n = 4$ independent differentiations. **i** Quantification of AF-synap uptake by flow cytometry for $n = 3$ independent differentiations. WT vs C71G$^{+/-}$ comparisons were not statistically significant (two-way ANOVA with Šídák's multiple comparisons test). Uptake with BafA pre-treatment (**$P = 0.0044$, two-way ANOVA with Šídák's multiple comparisons test, $t = 4.11$, df = 12 for +BafA vs −BafA). **j**–**l** Assay with 48 h post-washout. **j** Representative images of the residual pHrodo-synaptosome signal (red) merged with CD68 (green). Zoomed images of pHrodo-synaptosomes colocalizing with CD68. Scale bar: 10 μm and 2 μm. **k** Quantification of the fraction of CD68 volume occupied by pHrodo-synaptosome signal (yellow in **j**) normalized to WT iMGs (**$P = 0.0011$, $t = 8.401$, df = 4) for $n = 3$ independent differentiations. **l** Quantification of the percentage of cells retaining pHrodo signal ($n = 3$ differentiations) normalized to WT iMGs (*$P = 0.0490$, unpaired two-tailed $t$-test, $t = 2.462$, df = 6). All graphs show mean ± SEM, with each data point representing an independent differentiation. Additional statistical comparisons are in Data S1. Source data are provided as a Source Data file.

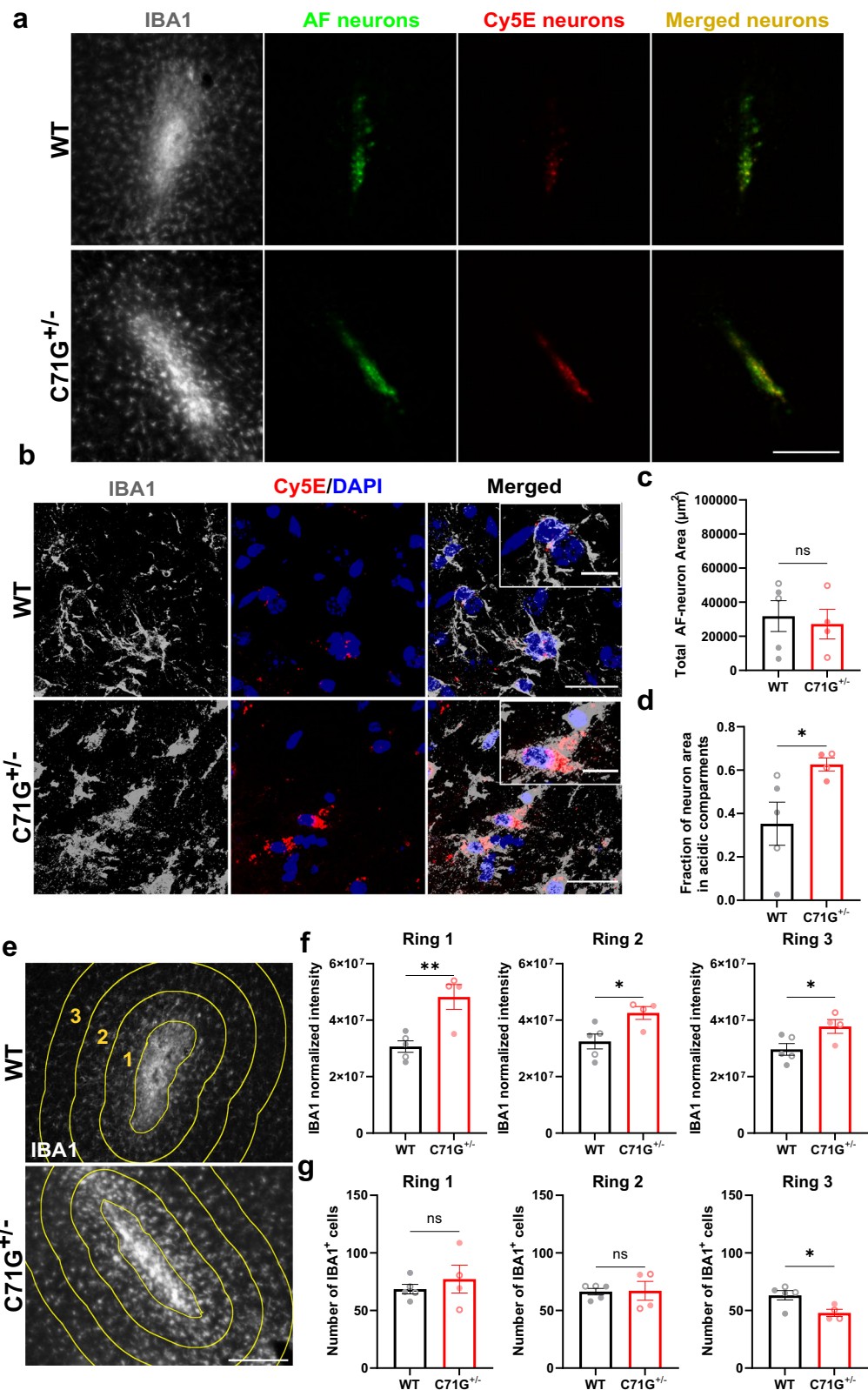

significantly higher volume of synaptosomes within CD68-positive compartments compared to WT iMGs (Fig. 4k). Further, a higher percentage of PFN1 C71G[+/-] iMGs retained pHrodo-labeled synaptosomes compared to control iMGs (Fig. 4l). Iba1 intensity changed in iMGs as a function of synaptosome phagocytosis, however there were no differences between PFN1 genotypes (Supplementary Fig. 10). Together, these data indicate that vesicular degradation of phagocytosed

material, rather than the uptake of material, is less efficient in ALS-PFN1 iMGs compared to controls.

To investigate whether deficient degradation of phagocytosed material occurs in in vivo models of ALS-PFN1, dead neurons were stereotactically injected into the motor cortex of PFN1 C71G[+/-] knock-in and WT mice (Fig. 5), as microglia are known to phagocytose dead neuron material within the mammalian brain[39]. PFN1 C71G[+/-] mice were

**Fig. 5 | Differential processing of dead neurons in ALS-PFN1 mouse brains compared to controls.** Analysis of PFN1 C71G[+/−] (n = 4) and WT (n = 5) mouse brains injected with dead neurons co-labeled with the constitutively fluorescent dye Alexa Fluor 546 (AF-neurons) and the pH-sensitive dye Cypher5E (Cy5E-neurons) 72 h post-injection. **a** Representative immunofluorescence images at the injection site in the mouse motor cortex of the myeloid marker IBA1(gray), the residual dead neurons (AF-neurons in green), and the dead neurons localized in acidic compartments (Cy5E-neurons in red) as well as the overlaid image of AF-neurons and Cy5E-neurons (merged in yellow). Scale bar: 200 μm. **b** Representative confocal images of IBA1-positive (IBA1[+]) cells colocalizing with Cy5E-neuron signal from a PFN1 WT (top) and PFN1 C71G[+/−] (bottom) mouse brain (scale bar: 25 μm) including higher magnification insets (scale bar: 10 μm). **c** Quantification of the total area of residual AF-neuron signal (ns P = 0.7288, t = 0.3610, df = 7). **d** The fraction of Cy5E-neuron (total area) signal that overlays with the AF-neuron (total area) signal from **c** (*P = 0.0499, t = 2.366, df = 7). **e** Representative IBA1 (gray) immunofluorescence images at the injection site for WT PFN1 and PFN1 C71G[+/−] mice. Three concentric rings (yellow) around the site of injection (center ring) are shown for analysis in (**f** and **g**). Scale bar: 100 μm. **f** Quantification of total IBA1 signal intensity in the three rings surrounding the injection site labeled in (**e**) normalized to the number of IBA[+] cells (**P = 0.0058, t = 3.909, df = 7 for ring 1; *P = 0.0264, t = 2.804, df = 7 for ring 2; *P = 0.0374, t = 2.563, df = 7 for ring 3). **g** Quantification of the number of IBA1[+] cells in each ring indicated in (**e**) (ns P = 0.4752, t = 0.7545, df = 7 for ring 1; ns P = 0.9264, t = 0.09573, df = 7 for ring 2; *P = 0.0255, t = 2.827, df = 7). Unpaired two-tailed t-test was used for all statistical comparisons. Graphs in this figure show mean ± SEM. Data points represent individual animals. Males are plotted open symbols and females with closed symbols. Source data are provided as a Source Data file.

generated with CRISPR/Cas9 technology such that the analogous C71G mutation was introduced into the endogenous mouse *PFN1* gene locus at one allele, resulting in physiological expression of PFN1 C71G. Total PFN1 protein levels were reduced in PFN1 C71G[+/−] mice compared to WT controls (Supplementary Fig. 12a, b) as observed in iMGs (Fig. 2). Consistent with other ALS and ALS/FTD mouse models lacking mutant protein overexpression[40], overt phenotypes related to motor neuron dysfunction (Supplementary Fig. 12c, d) and gliosis were not observed in PFN1 C71G[+/−] mice aged to ~600 days (Supplementary Fig. 13). Nevertheless, PFN1 C71G[+/−] mice provide an in vivo system to study the effects of PFN1 mutation on phagocytic processing. To this end, dead neurons were co-labeled with the constitutively fluorescent dye Alexa Fluor 546 (AF) to measure the total amount of material present in the brain tissue and the pH-sensitive dye Cypher5E (Cy5E) to quantify material that is specifically located within acidic phagosomal compartments (Fig. 5a, b). Seventy-two hours post-injection, a range of AF signal was detected in the mouse brains from both genotypes; this range likely reflects some variability in the amount of dead neuron material that effectively diffused within the tissue post-injection (Fig. 5c). Notably, a significantly higher fraction of the dead neuron material was retained in acidic compartments of PFN1 C71G[+/−] mouse tissue compared to WT littermates (Fig. 5a, d). Further, brain sections from PFN1 C71G[+/−] mice that contained dead neurons exhibited higher IBA1 signals (Fig. 5e, f). This was not due to more IBA1-positive cells, which were similar in number between genotypes except for lower numbers in the mutant mouse tissue represented by Ring 3 (Fig. 5g); we speculate that migration of microglia from the region within Ring 3 towards the site of dead neurons accounts for this difference. These data are consistent with heightened microglial reactivity and enhanced IBA1 expression in response to the presence of dead neurons in PFN1 C71G[+/−] mice. Although this study was not powered to detect sex differences, both males and females were included (distinct symbols are used for males versus females in all graphs and segregated data is included in the Source Data file); data generated from mice do not appear to be confounded by sex. Collectively, these results support the notion that expression of ALS-linked PFN1 in phagocytic cells impairs degradation of engulfed substrates both in vitro and in vivo.

### Lysosomes accumulate in the perinuclear region of ALS-PFN1

As lysosomes represent a terminal organelle in the endo-lysosomal pathway through which cargo is degraded during phagocytosis, endocytosis, and autophagy, we probed for evidence of lysosomal dysfunction in ALS-PFN1 iMGs. The quenched-bovine serum albumin (DQ-BSA) assay was used to measure lysosomal degradative capacity in iMGs. The DQ-BSA substrate is internalized through pinocytosis, and subsequent proteolysis by lysosomal proteases results in a fluorescence signal[41]. No significant differences in DQ-BSA signal intensity were observed between PFN1 C71G[+/−] or M114T[+/−] iMGs and WT controls (Fig. 6a, b). The pH of intracellular acidic compartments within iMGs was also examined with Lysosensor DND-189. As expected, pretreating iMGs with BafA resulted in a decrease in fluorescence,

consistent with an increase in lysosomal pH (Fig. 6c). However, there was no difference in DND-189 intensity between ALS-PFN1 iMGs and WT controls. Further, similar levels of both the lysosomal-associated membrane protein 1 (LAMP1) and the mature lysosomal hydrolase Cathepsin D (mCTSD) were detected in PFN1 C71G[+/−] and WT iMGs by Western blot analysis (Fig. 6d–f). Collectively, these data argue against inherent defects in lysosome function in ALS-PFN1 iMGs. Strikingly, perinuclear clustering of LAMP1-positive lysosomes was observed in approximately twice as many ALS-PFN1 iMGs than WT iMGs (Fig. 6g, h). Indeed, excessive perinuclear clustering is observed under conditions of delayed phagosome maturation[42] and blocked autophagy[43].

### Rapamycin ameliorates phagocytic deficits in ALS-PFN1 iMGs

To further investigate perturbations in the autophagy pathway, lipidation of the ATG8 family protein microtubule-associated protein 1A/1B-light chain 3B (MAP1LC3B or LC3) was assessed by Western blot analysis. Lipidation entails the conversion from LC3I to LC3II, where LC3II binds to autophagosome membranes during autophagy initiation. There did not appear to be a defect in autophagosome formation in PFN1 C71G[+/−] iMGs, as the ratio of LC3II to LC3I (LC3II/LC3I) and total LC3 levels were similar to WT iMGs (Fig. 7a–c). However, the autophagy receptor p62/SQSTM1 (sequestosome-1) was significantly elevated in the PFN1 C71G[+/−] iMGs compared to WT iMGs at baseline. Bafilomycin A treatment led to an overall increase in p62 levels, but without a significant difference between genotypes, indicative of reduced autophagic processing in untreated PFN1 C71G[+/−] iMGs (Fig. 7d–e). Immunofluorescence staining for LC3 and p62 confirmed these results by revealing similar abundance and size of LC3-positive vesicles and an increase in abundance and size of p62-positive puncta in PFN1 C71G[+/−] iMGs compared to WT (Fig. 7 f–j). A Pearson's analysis showed greater colocalization of p62-positive and LC3-positive puncta in PFN1 C71G[+/−] iMGs, which could indicate that LC3-positive vesicles have accumulated p62 but not undergone processing of cargo in PFN1 C71G[+/−] iMGs. We also considered whether early endosome antigen 1 (EEA1)-positive endosomes were altered in PFN1 C71G[+/−] iMGs, as EEA1-positive endosomes are normally cleared through the autophagy pathway[44]. Indeed, immunofluorescence analysis revealed an increase in EEA1 signal in PFN1 C71G[+/−] iMGs compared to WT, which could indicate a delay in fusion of endosomes with autophagosomes and/or phagosomes (Supplementary Fig. 14a, b).

Considering that autophagy and phagocytosis pathways appear perturbed by expression of mutant PFN1, we administered rapamycin, a potent inhibitor of the mammalian target of rapamycin (mTOR) that leads to activation of autophagy[45]. Autophagic induction by rapamycin was also shown to stimulate clearance of phagocytosed *M. tuberculosis* by overriding the inhibition of this pathogen on phagosomal maturation[46]. Here, administration of rapamycin to iMGs prior to the phagocytosis assay with pHrodo-labeled mouse synaptosomes prevented the accumulation of pHrodo signal in ALS-PFN1 iMGs, normalizing the phagocytosis index in mutant iMGs to that of WT iMGs (Fig. 8a, b and Supplementary Fig. 14c). Further, rapamycin

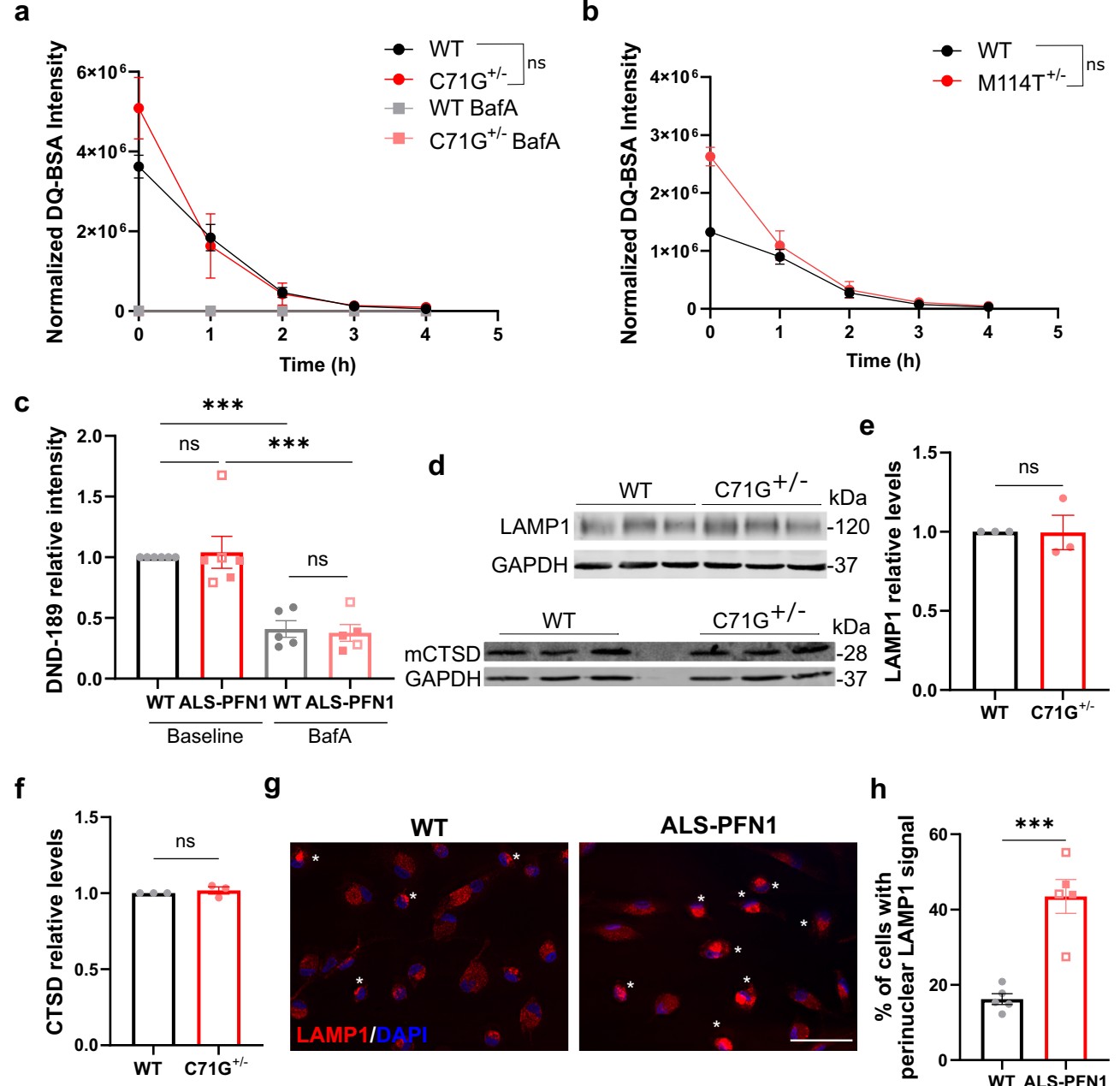

**Fig. 6 | Lysosomal perinuclear accumulation occurs in ALS-PFN1 iMGs. a, b** De-quenched DQ-BSA fluorescence intensity measured over 4 h by live-cell imaging of PFN1 WT and C71G$^{+/-}$ iMGs (**a** ns $P = 0.5740$, F (1, 4) = 0.3737) and of PFN1 WT and M114T$^{+/-}$ iMGs (**b** ns $P = 0.0717$, F (1, 4) = 5.921). Data were normalized to total cell number. Statistics were obtained by two-way ANOVA for $n = 3$ independent differentiations. **c** The fluorescence intensity of Lysosensor DND-189 was quantified for WT ($n = 6$ for two different lines) and ALS-PFN1 (C71G$^{+/-}$ $n = 3$, solid squares and M114T$^{+/-}$ $n = 3$, open squares) iMGs at baseline or pre-treated with 200 nM Bafilomycin A (WT $n = 5$ from two different lines and ALS-PFN1 from C71G$^{+/-}$ $n = 3$, solid squares and M114T$^{+/-}$ $n = 2$, open squares, BafA; ***$P = 0.0006$ for WT untreated vs BafA or $P = 0.0002$ for ALS-PFN1 untreated vs BafA; ns $P = 0.9840$ for untreated WT vs ALS-PFN1 or $P = 0.9938$ for BafA WT vs ALS-PFN1). Data was normalized to WT iMGs for each independent differentiation. Statistics were determined by one-way ANOVA (F 0.1243 = (3, 18)) and Tukey's multiple comparisons test. **d** Western blot

analysis of LAMP1 and the mature form of Cathepsin D (mCTSD) from WT and C71G$^{+/-}$ ($n = 3$ independent differentiations) with GAPDH as a loading control. **e, f** Quantification of (**d**) for LAMP1 (**e** ns $P = 0.9703$, $t = 0.03967$, df = 4) and mCTSD (**f** ns $P = 0.4999$, $t = 0.7409$, df = 4). Levels of the target proteins were first normalized to GAPDH and then to the WT control lane corresponding to the same differentiation. Statistics were determined by unpaired two-tailed $t$-test. **g** Representative immunofluorescence images of LAMP1 in PFN1 WT ($n = 7$) and ALS-PFN1 (C71G$^{+/-}$ $n = 4$ and M114T$^{+/-}$ $n = 3$) iMGs. White asterisks indicate cells with perinuclear LAMP1 signal. Scale bar = 50 μm. **h** Percentage of iMGs showing LAMP1 perinuclear localization for WT ($n = 5$ for two iPSC lines) and ALS-PFN1 (C71G$^{+/-}$ $n = 2$, solid squares and M114T$^{+/-}$ $n = 3$, open squares) iMGs. Statistics were determined by unpaired two-tailed $t$-test (***$P = 0.0004$, $t = 5.768$, df = 8). All graphs show mean ± SEM and individual data points in the bar graphs represent independent differentiations. Source data are provided as a Source Data file.

restored p62 levels in PFN1 C71G$^{+/-}$ iMGs to that of WT as indicated by Western blot analysis (Fig. 7d, e). The effects of rapamycin lend further support to the notion that mutant PFN1 impairs vesicular processing in iMGs[47].

## ALS-linked PFN1 exhibits enhanced binding to PI3P
We next sought to explore interactions between PFN1 and factors involved in vesicular degradation and considered lipid interactions given that PFN1 binds phosphoinositides[12,48]. A particular

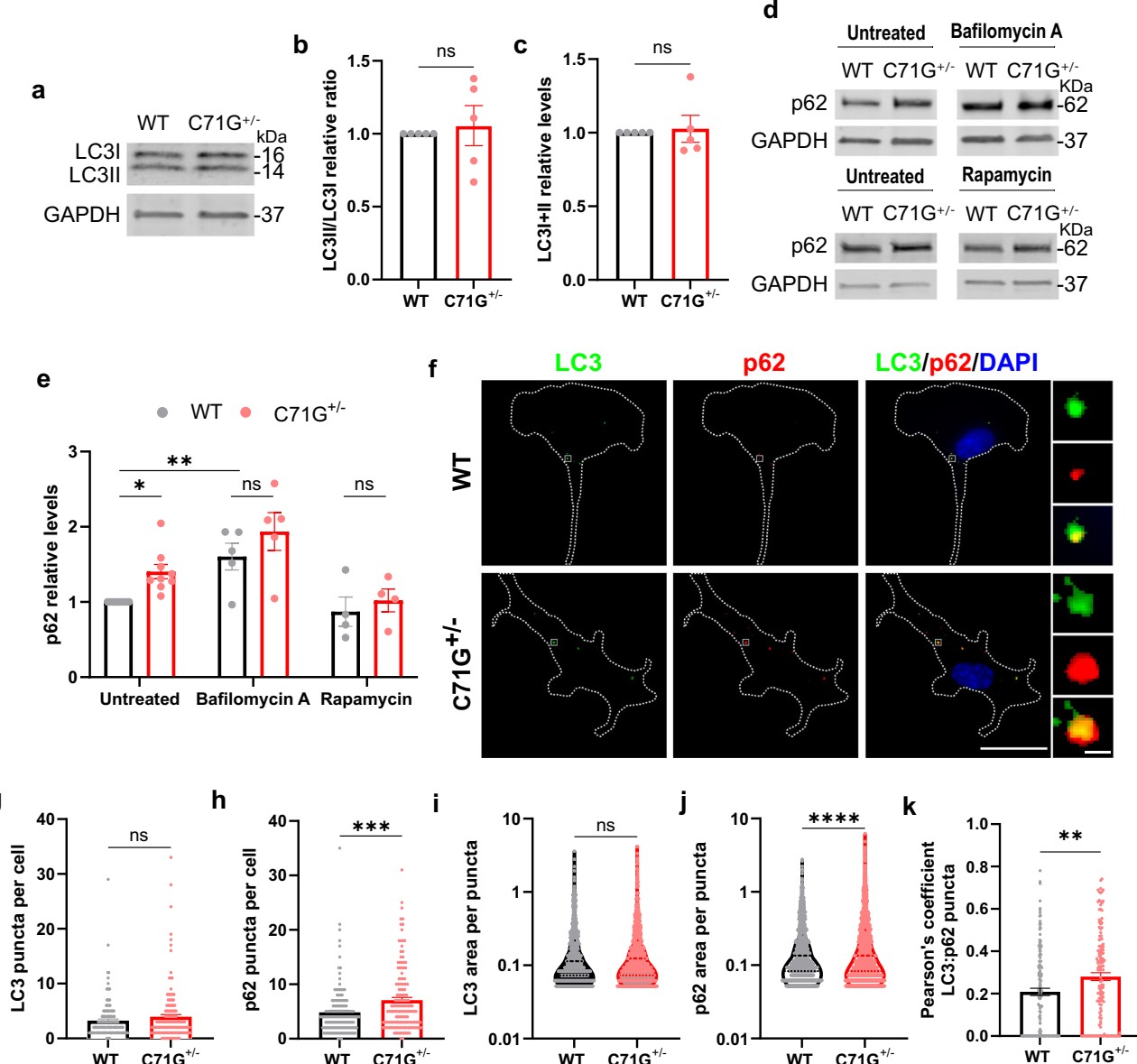

**Fig. 7 | ALS-PFN1 iMGs show deficits in autophagy-related proteins. a–c** LC3I and LC3II expression in PFN1 iMGs by Western blot analysis from $n = 5$ independent differentiations. **a** Representative Western blot of LC3I, LC3II, and GAPDH as loading control. **b, c** Quantification of LC3I and LC3II in (**a**). **b** LC3II/LC3I ratio normalized to WT iMGs in each differentiation (ns $P = 0.6978$, $t = 0.403$, df = 8). **c** Total LC3 levels (LC3I + LC3II) normalized to GAPDH and reported as relative to WT iMGs (ns $P = 0.7810$, t = 0.288, df = 8). **d–e** p62 protein levels in PFN1 iMGs after bafilomycin A or rapamycin treatment compared to untreated cells. **d** Representative Western blots of p62 for PFN1 iMGs under untreated conditions or 0.1 μM rapamycin ($n = 4$ independent differentiations) and under untreated conditions or 100 nM bafilomycin A ($n = 5$ independent differentiations). **e** Quantification of (**d**), with p62 bands normalized to GAPDH and all data reported as relative to untreated WT iMGs. (Untreated WT vs C71G+/− *$P = 0.0430$, $t = 2.594$, df = 30; rapamycin WT vs C71G+/− ns $P = 0.8917$, $t = 0.646$, df = 30; bafilomycin A WT

vs C71G+/− ns $P = 0.3200$, $t = 1.597$, df = 30). Statistics were determined by two-way ANOVA and Šídák's multiple comparisons test. **f** Representative immuno-fluorescence images of LC3 (green) and p62 (red) puncta with DAPI (blue) in PFN1 iMGs. Scale bar: 25 μm and 1 μm (inset). **g, h** Quantification of puncta per cell for LC3 (**g** ns $P = 0.1689$, $t = 1.379$, df = 289) and p62 (**h** ***$P = 0.0002$, $t = 3.748$, df = 289) from $n = 147$ WT and 144 C71G+/− cells across 4 independent differentiations. **i, j** Quantification of the puncta size for LC3 (**i** ns $P = 0.1828$, $t = 1.332$, df = 2434) and p62 (**j** ****$P < 0.0001$, $t = 4.509$, df = 3660) from $n = 960$ WT and 1476 C71G+/− puncta from the same cells as (**g, h**). **k** Pearson's coefficient of the colocalization between LC3 and p62 puncta (**$P = 0.0040$, $t = 2.898$, df = 285) from $n = 144$ WT and 143 C71G+/− cells across 4 independent differentiations. All bar graphs show mean ± SEM and data points represent individual differentiations (**b, c, e**), cells (**g, h, k**), or puncta (**i, j**). Statistics for bar graphs other than **e** were determined by unpaired two-tailed $t$-test. Source data are provided as a Source Data file.

phosphoinositide of interest, phosphatidylinositol 3-phosphate (PI3P), functions as a critical signaling lipid in autophagy and endocytic processing through binding interactions with FYVE-finger proteins, which was highlighted in the proteomics pathway analysis (Fig. 2e)[20,21]. To examine whether PFN1 binds PI3P, we first used a differential scanning fluorimetry (DSF) assay that reports on PFN1 melting temperature ($T_m$), a measure of protein stability, that we used before to study PFN1 binding with other ligands[29]. In the absence of ligand, the $T_m$ of PFN1

M114T and C71G is, respectively, -14 °C (Fig. 9a) and -20 °C (Supplementary Fig. 15a), below that of PFN1 WT, consistent with a destabilizing effect of these mutations on PFN1 with C71G being more severe[29]. Increasing concentrations of PI3P caused a reduction in the $T_m$ of all PFN1 proteins (Fig. 9a–c). Upon addition of 200 μM PI3P, PFN1 M114T exhibited a significantly larger decrease in $T_m$ ($\Delta T_m$) than PFN1 WT (Fig. 9a–c) and C71G unfolded to the extent that a proper binding curve was not observed (Supplementary Fig. 15a). Due to the severe

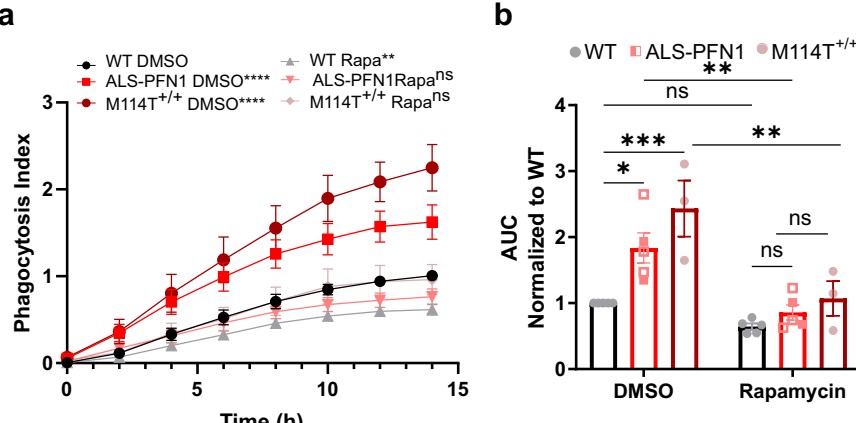

**Fig. 8 | Rapamycin ameliorates deficits in phagocytosis in ALS-PFN1 iMGs.** Live-cell phagocytosis assays using pHrodo-labeled mouse synaptosomes as substrate for WT ($n = 5$ for two different lines), ALS-PFN1 (C71G[+/-] $n = 2$ and M114T[+/-] $n = 3$), and M114T[+/+] ($n = 3$) iMGs pre-treated with 0.1 μM rapamycin. **a** Quantification of the phagocytosis index (see "Methods" section). Two-way ANOVA and Tukey's multiple comparisons test was used for statistical analysis. Comparisons relative to WT iMGs with DMSO are shown (****$P < 0.0001$, **$P = 0.0015$, ns $P = 0.4638$ for ALS-PFN1 with 0.1 μM Rapamycin, and ns $P > 0.9999$ for M114T[+/+] with 0.1 μM Rapamycin). **b** Area under the curve (AUC) determined from (**a**). Within the ALS-PFN1 group, solid squares are for C71G[+/-] $n = 2$, open squares are for M114T[+/-] $n = 3$. Statistics were determined by two-way ANOVA and Šídák's multiple comparisons test for: the DMSO condition (*$P = 0.0326$ for WT vs ALS-PFN1 and ***$P = 0.0006$ for WT vs M114T[+/+]), the DMSO vs 0.1 μM Rapamycin condition per genotype ($P = 0.9152$ for WT, **$P = 0.0081$ for ALS-PFN1, and **$P = 0.0038$ for M114T[+/+]) and the 0.1 μM Rapamycin condition ($P = 0.9993$ for WT vs ALS- PFN1, $P = 0.8904$ for WT vs M114T[+/+]). All graphs show mean ± SEM and individual data points in the bar graphs represent independent differentiations. Source data are provided as a Source Data file.

instability of PFN1 C71G in vitro, subsequent experiments were pursued with PFN1 M114T. Similar effects on the $T_m$ of PFN1 WT and M114T were observed with the plasma-membrane localized 4,5-bisphosphate (PI(4,5)P2) (Supplementary Fig. 15b, c), a well-characterized PFN1 ligand[48]. Intriguingly, the effect of PIP binding to PFN1 had the opposite effect as poly-proline ligands, which stabilized PFN1 (i.e., lead to an increase in $T_m$)[27,29]. The biological reason for PIP-induced destabilization of PFN1 remains to be explored but could be relevant to the role of PIPs in cellular signaling.

We turned to Nuclear Magnetic Resonance (NMR) spectroscopy to further examine the binding interaction between PFN1 variants and the autophagy-associated PI3P. $^{15}$N-$^1$H heteronuclear single-quantum coherence (HSQC) spectra were collected with either $^{15}$N labeled PFN1 WT (Fig. 9d) or M114T (Fig. 9e) as a function of increasing PI3P concentration. PFN1 chemical shift perturbations, indicative of PI3P binding, were quantified (Supplementary Fig. 15d) and mapped onto the structure of PFN1 WT in complex with actin and a poly-proline sequence from vasodilator-stimulated phosphoprotein (VASP)[49]; the crystal structure of this ternary complex was used to place chemical shift perturbations arising from PI3P into context with other PFN1 ligand binding sites. PFN1 residues that show major chemical shift perturbations in response to PI3P binding are located at or near poly-proline binding sites and include residues within the N- and C-terminal α-helices (α₁ and α₄) and the β-strands behind them (β₁, β₂, and β₇) (Supplementary Fig. 15d, e). Overall, larger chemical shift perturbations were observed in PFN1 WT (Fig. 9f) than in PFN1 M114T (Fig. 9g) upon titration of PI3P. A global fitting analysis of residues with robust PI3P-induced chemical shift perturbations was performed (Supplementary Fig. 16), resulting in an apparent dissociation constant ($K_d$) of 1880 ± 190 μM for PFN1 WT and 370 ± 30 μM for PFN1 M114T with respect to PI3P binding. The smaller chemical shift perturbations together with a 5-fold higher binding affinity indicates that PFN1 M114T undergoes smaller conformational changes to accommodate tighter PI3P binding. We also observed precipitation of PFN1 M114T starting at 85 μM PI3P during the titration experiment, whereas PFN1 WT remained soluble with PI3P concentration up to 1.1 mM. These observations are consistent with reduced PFN1 stability resulting from both the M114T mutation and binding to PI3P (Fig. 9a–c).

## Discussion

The mechanism by which mutations in PFN1 cause ALS is unknown. Prior reports with rodent models showed that ALS-linked mutations in PFN1 induce motor neuron degeneration[50,51]. However, these studies were carried out under conditions of PFN1 overexpression and did not address the effects of ALS-PFN1 on microglia. While the core functions of microglia are needed to develop and maintain a healthy CNS[1], microglia dysfunction contributes to neuron degeneration through non-cell-autonomous mechanisms[2]. In support of this notion, motor neuron degeneration is modulated by ALS gene expression in microglia[52]. Studies with iPSC-derived microglia cells are providing insights into how disease-linked genes alter the intrinsic properties of human microglia[8,53,54]. Though it is widely recognized that the properties of microglia differ between in vitro and in vivo environments[55], emerging evidence supports that iMGs are a relevant model for investigating the effects of disease-linked genes on canonical microglia functions[7]. Herein, immunofluorescence, transcriptomics, LPS, and phagocytosis studies demonstrate a commitment of ALS-PFN1 and WT iMGs toward a microglia cell fate. Our experiments also revealed that expression of mutant PFN1 disrupts phagocytic processing, a core microglia function. Parallel studies in knock-in PFN1 C71G[+/-] mice intracranially injected with dead neurons recapitulated a defect in phagocytic processing, demonstrating consistency between disease-relevant phenotypes in iMGs and in vivo.

Our longitudinal and immunofluorescence analyses showed deficient processing of phagocytosed material within ALS-PFN1 iMGs relative to controls. Evidence from multiple lysosomal functional assays indicate that defective phagocytic processing in ALS-PFN1 iMGs is not caused by reduced lysosomal degradative capacity. Rather, the accumulation of phagocytosed material likely stems from inefficient vesicular processing through the endo-lysosomal pathway. Intriguingly, perinuclear accumulation of lysosomes was observed in ALS-PFN1 iMGs, a condition that has been observed in other neurological disorders[56]. Lysosome positioning is dependent on cytoskeletal dynamics, which are perturbed by ALS-linked mutations in PFN1[27,57]. Although ALS-PFN1 iMGs did not exhibit deficiencies in F-actin levels or in the uptake of synaptosomes, both of which require actin polymerization, we cannot exclude the possibility that subtle alterations in

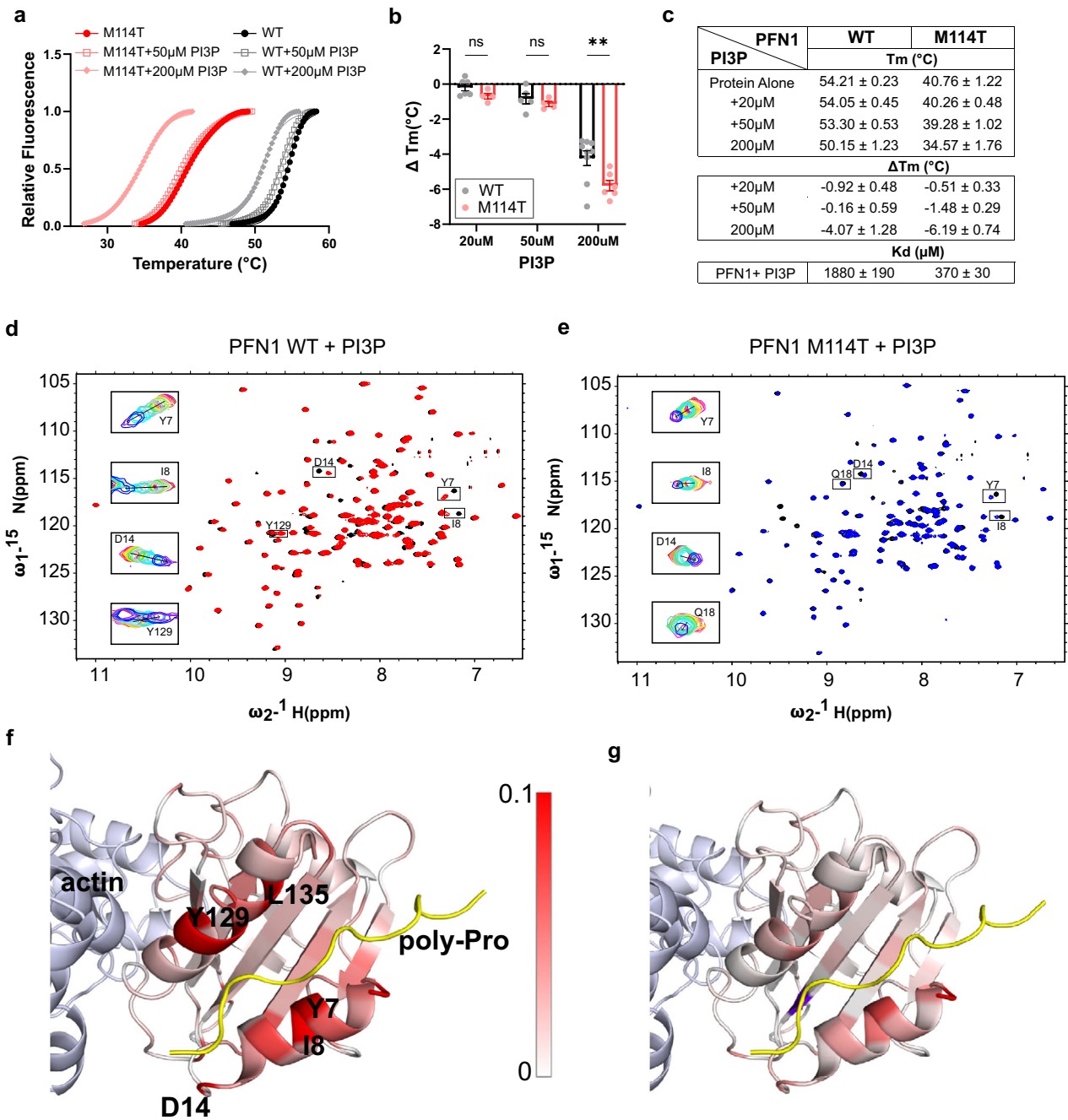

**Fig. 9 | PI3P is a PFN1 ligand with enhanced binding affinity for PFN1 M114T.**
**a**, **b** Differential scanning fluorimetry (DSF) with PFN1 WT and M114T proteins in the presence of PI3P. **a** Thermal denaturation profiles of PFN1 proteins incubated with PI3P measured by SYPRO Orange fluorescence as a function of increasing temperature. An average of two technical replicates is shown and is representative of $n = 4-9$ independent experiments. The curves were fit to the Boltzmann's sigmoidal function to determine the apparent melting temperature ($T_m$). **b** $\Delta T_m$ reports the difference between the $T_m$ at the indicated PI3P concentration and the protein without PI3P. Statistics were determined using two-way ANOVA F $(1, 28) = 7.115$ and Šídák's multiple comparisons test (**$P = 0.0038$ for 200 μM and ns $P = 0.735$ for 50 μM or 0.9326 for 20 μM) for WT $n = 9$ and M114T $n = 8$ independent experiments using different PI3P concentrations as described in the Source Data file. Bar graphs show mean ± SEM with each data point representing an independent experiment.

**c** Summary of $T_m$ and $\Delta T_m$ obtained from (**b**) and the dissociation constants from the NMR studies (**d**–**g**) for PFN1 WT and M114T with PI3P. **d**–**g** Titration of PFN1 with PI3P using NMR. **d** Overlay of $^{15}$N-$^1$H HSQC spectra of PFN1 free (black) and bound to PI3P (red) for PFN1 WT. **e** the same as (**d**) except for PFN1 M114T in the free state (black) and bound to PI3P (blue). **d**, **e** Insets show the overlay of spectra collected during the titration of PI3P for select residues. **f**, **g** Chemical shift differences between spectra of PFN1 alone and the PI3P-PFN1 complex are mapped onto the structure of PFN1 in the ternary complex with actin (light blue) and PLP (yellow); pdb ID 2PAV for PFN1 WT (**f**) and PFN1 M114T PFN1 (**g**). Residues are colored according to the chemical shift perturbation measured upon PI3P binding as indicated by the scale bar, with white and red corresponding to 0–0.1 ppm, respectively. PFN1 residues presenting chemical shift perturbations ≥0.1 ppm are shown in red. Additional information is in Supplementary Fig. 15. Source data are provided as a Source Data file.

cytoskeletal dynamics contribute to inefficient vesicular processing in ALS-PFN1 iMGs.

Lysosome positioning is important for both phagocytosis and autophagy[42,43], the two prominent vesicular degradation pathways in

microglia (Fig. 10)[47]. While phagocytosis and autophagy represent distinct pathways, these pathways can both utilize endosomes for effective vesicular processing and ultimately converge on lysosomal fusion for degradation of cargo. Further, autophagy and phagocytosis

converge through a process called LC3-associated phagocytosis (LAP)[47,58]. During LAP, LC3 is recruited to phagosomes where it can interact with parts of the autophagic machinery to mediate more efficient clearance of the phagosomal cargo[47]. Our studies did not uncover differences in conversion of LC3I to LC3II, a process that is required for LAP and autophagy[58]. Further, our LC3 immunofluorescence analysis revealed similar numbers of LC3 puncta between mutant and WT iMGs, indicating that mutant PFN1 does not interfere with the formation of LC3-decorated vesicles. However, multiple observations are consistent with reduced autophagic processing, including elevated p62 protein observed by Western blot and elevated p62 puncta detected through immunofluorescence analyses, where more p62 was also detected within LC3/p62-containing puncta in mutant PFN1 iMGs. Further, elevation of TBC1D15 and RASD2 at both the mRNA and protein level are consistent with an attempt by mutant PFN1 iMGs to upregulate autophagy, as previous studies have shown a positive correlation between TBC1D15 and RASD2 expression and autophagy[33,34]. Specifically, both TBC1D15 and RASD2 function in mitophagy, a form of selective autophagy that removes old or damaged mitochondria[33,34]. While it remains to be determined if mitochondria are dysfunctional and/or if mitophagy is altered in mutant PFN1 iMGs[28], enhanced association of TBC1D15 with mutant PFN1 mitochondria indicates these organelles are marked for degradation via the autophagy pathway. Notably, knockdown of PFN1 in HMC3 cells resulted in a downregulation of TBC1D15, the opposite effect as mutant PFN1 expression in iMGs, suggesting the latter is due to a gain-of-toxic mutant PFN1 function.

That lipid droplets accumulate within mutant PFN1 iMGs is also indicative of deficient autophagic processing in mutant PFN1 iMGs. Lipid droplets are processed by a form of selective autophagy called lipophagy that relies on autophagosome-mediated degradation of lipids (Fig. 10)[59]. Microglia/macrophages containing excessive lipid droplets exhibit compromised phagosome maturation and enhanced

cytokine release[32,60]. Indeed, our observations in mutant iMGs are consistent with deficient phagosome maturation, which can explain, at least in part, why phagocytosed material is inefficiently processed through the endo-lysosomal pathway. In contrast to iMGs harboring ALS/FTD-linked mutations in C9ORF72[8], mutant PFN1 iMGs did not exhibit substantial differences in cytokine release at baseline or when challenged with LPS. It is possible that the inflammatory response of microglia expressing ALS-linked PFN1 is altered by conditions not tested herein, such as in vivo under conditions where the CNS is challenged by chronic accumulation of debris. Expanding our iMG studies into a co-culture format with neurons, as well as further investigations with our knock-in PFN1 C71G[+/−] mouse model may reveal additional phenotypes induced by ALS-PFN1[6]. Interestingly, glial cells buffer lipids from neighboring neurons under conditions of neuronal stress in vivo[61]. It will be important to determine whether lipid droplet accumulation and lipid dysmetabolism in glia preclude this neuro-protective function, which could have implications for neurodegeneration. In the context of mutant PFN1 iMGs, lipid droplet accumulation appears to result from a gain-of-toxic mutant PFN1 function, as PFN1 knockdown in HMC3 cells failed to recapitulate this phenotype.

Other factors that play central and key functions in both phagocytosis and autophagy are phosphoinositides, signaling lipids that associate with cellular and vesicular membranes to coordinate the recruitment of effector proteins[21]. In particular, PI3P is a critical regulator of autophagy, being required throughout the autophagic pathway for autophagy initiation, autophagosome maturation, and effective fusion of autophagosomes with lysosomes[20]. PI3P is also involved in LAP, phagosome maturation, and endo-lysosomal processing[20,21]. PI3P binds proteins containing Phox homology and FYVE-finger domains, recruiting these effector proteins to vesicular membranes during autophagy and/or phagocytosis. Herein, we demonstrate a novel binding interaction between PFN1 and PI3P,

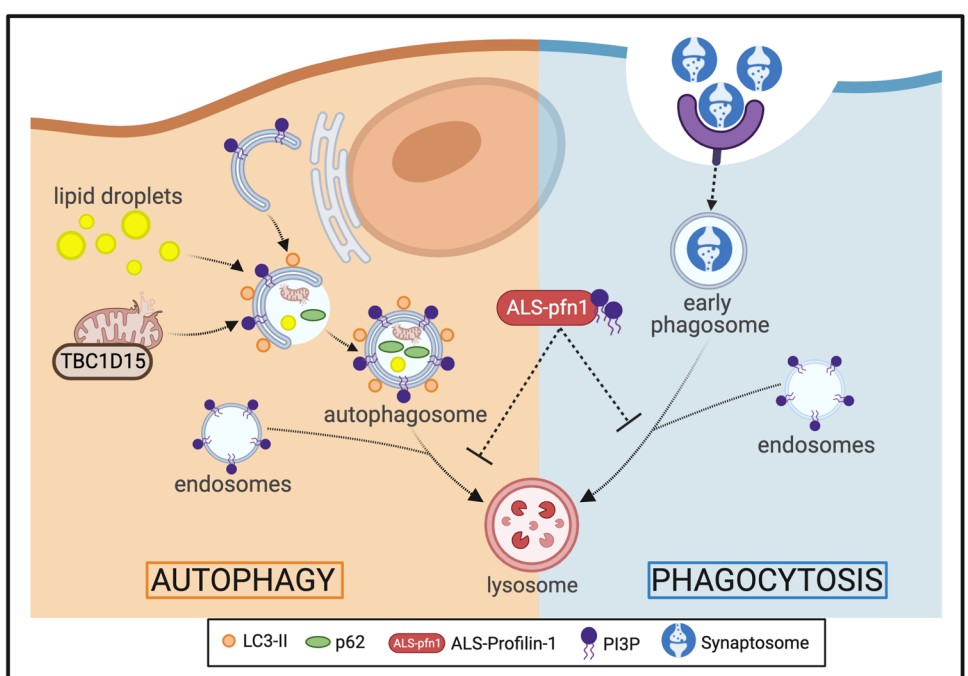

**Fig. 10 | Proposed model for impaired vesicular degradation in mutant PFN1 iMGs.** Mutant iMGs are capable of engulfing substrate (phagocytosis pathway) and forming autophagosomes (autophagy pathway). However, mutant PFN1 expression impairs degradative processing through these pathways, accounting for the accumulation of lipid droplets, upregulation of TBC1D15 at mitochondria, and delayed degradation of phagocytosed material. Vesicles within these pathways contain PI3P, a signaling lipid that is central and critical for efficient vesicular maturation and degradation. Mutant and misfolded PFN1 exhibits enhanced binding to PI3P, raising the possibility that mutant PFN1 interferes with endo-lysosomal processing through a gain-of-toxic function involving PI3P signaling. Created with BioRender.com.

where mutant PFN1 M114T exhibits ~5-fold enhanced binding affinity for this signaling lipid. Based on these observations, we propose a model whereby mutant and misfolded PFN1 engages in aberrant interactions with PI3P within the autophagy and phagocytosis pathways, leading to inefficient vesicular degradation of both intracellular and phagocytosed material. Mutant PFN1-induced deficits were ameliorated upon administration of rapamycin, which is known to stimulate autophagy as well as the maturation of phagosomes into phagolysosomes[46]. Therefore, rapamycin likely enhances vesicular degradation in mutant PFN1 iMGs by acting upon both autophagy and phagocytosis, as evidenced by normalization of p62 and the phagocytic index to WT iMG levels, respectively. It will be of interest to further investigate the PFN1/PI3P interaction in context of vesicular degradation in cells, and to define the precise vesicles and steps within this process that are most vulnerable to mutant PFN1 expression. To date, we have not been able to modulate gene expression in iMGs, as viral transduction was highly toxic to these cells. Recent advances with lentiviral approaches for modulating gene expression in iMGs should facilitate future studies aimed at manipulating endo-lysosomal and autophagy pathway components in ALS-linked PFN1 iMGs[7].

Our results have implications for ALS and related neurodegenerative disorders. In the context of ALS-PFN1, the outcomes of our study raise the possibility that microglial dysfunction caused by mutant PFN1 could directly affect the CNS microenvironment, and by extension the health and viability of motor neurons. Intrinsic microglia dysfunction may be more prevalent in ALS than previously thought, as other ALS-associated genes including *C9ORF72* are abundantly expressed in microglia. Further, multiple ALS-linked proteins, including tank-binding kinase (TBK1), optineurin (OPTN), ubiquilin 2 (UBQLN2), C9ORF72, and p62 are expressed in microglia and are directly involved with autophagy[62]. Indeed, pivotal roles for endo-lysosomal vesicular degradation pathways in ALS and ALS/FTD pathogenesis are emerging[22]. While there has been an emphasis on the autophagy pathway in neuronal subtypes across different neurodegenerative diseases, targeting canonical and non-canonical autophagy pathways in microglia may also be a viable therapeutic strategy for these disorders.

## Methods

### Animals

All mice were maintained in a standard barrier facility at UMass Chan Medical School. The Institutional Animal Care and Use Committee (IACUC) at UMass Chan Medical School approved all experiments involving animals. Standard breeding techniques were used for the generation of wild-type *C57BL/6J* embryos or pups for purification of synaptosomes or isolation of primary cortical neurons, respectively, as described below. *C57BL/6J-Pfn1^em4Lutzy/J* mice (#030313) were generated by and obtained from the Jackson Laboratory (Bar Harbor, ME) and referred to herein as PFN1 C71G$^{+/-}$ mice. These mice were CRISPR-engineered to harbor the allele *Pfn1^C71G, R75R*. The cohort of mice used in this study was generated by in vitro fertilization at the UMass Chan Medical School Transgenic Animal Modeling Core (TAMC; Worcester, MA) using male PFN1 C71G$^{+/-}$ mice and female wild-type *C57BL/6J* mice. Both males and females were included in all experiments with mice, although the study was not powered to detect sex differences.

### CRISPR engineering of PFN1 mutant iPSC lines

The PFN1 C71G and M114T lines and their respective isogenic control lines were generated from the euploid (46; XY) KOLF2.1J iPSC line[23] at The Jackson Laboratory for Genomic Medicine (Farmington, CT). The PFN1 C71G single-nucleotide variant T > G was generated using a guide RNA that overlaps codon 71 of PFN1 while the PFN1 M114T single-nucleotide variant T > C was generated with a guide RNA that overlaps codon 114 as listed below. A 100-nt single-stranded oligonucleotide donor template was used for both mutations as described before with

minor modifications[63]. The Amaxa 4D nucleofection system (Lonza) was used as follows: pre-assembled Cas9 RNP (2 μg IDT HiFi V3 Cas9 and 1.6 μg Synthego synthetic modified single guide RNA) and 40 pmol single-stranded oligonucleotide (IDT; HDR Donor oligo) were added to 20 μL P3 buffer containing $1.6 \times 10^5$ Accutase-dissociated iPSCs and the nucleofection procedure was carried out in a 16-well Amaxa 4D cuvette (Lonza; Primary Cell P3, pulse code CA137). Cells were cultured for 3 days under cold-shock conditions (32 °C/5% $CO_2$) and in the presence of RevitaCell (ThermoFisher) and HDR enhancer (1 μM Alt-R HDR Enhancer v2, IDT) for the first 24 h. Once the cells were confluent, iPSCs were dissociated and plated at low density to obtain single-cell-derived clones. Clones were screened by Sanger sequencing of PCR amplicons to identify the desired homology-directed repair events in PFN1 using the primers below.

For the PFN1 C71G variant:
Cas9 guide RNA: CCCGGATCACCGAACATTTC
Repair oligo:
TCTTGGTACGAAGATCCATGCTAAATTCCCCATCCTGCAGCAG
TGAGTCCCGGATCACCGAACCTTTCTGGCCCCCAAGTGTCAGCCCA
TTCACGTAAAA
PCR primer (forward): 5′- GAATCTTGGTGCACTGACTAACTTG
and PCR primer (reverse): 5′- AAGAACTCAAACGATGAACTCGATG
Sanger sequencing primer: 5′- ACTAACTTGATGGGCGCTTG
For M114T variant:
Cas9 guide RNA: CCAGCGCTAGTCCTGCTGAT
Repair oligo:
GGTGGGAGGCCATTTCATAACATTTCTTGTTGATCAAACCACCG
TGGACACCTTCTTTGCCCGTCAGCAGGACTAGCGCTGGAGGAGGAGG
AAAGAGAAA
PCR primer (forward): 5′-CACATGAATCAACAGTCTATGAGCC and PCR primer (reverse): 5′-CACACAGCACCTTGTTAGTAGAATC
Sanger sequencing primer: 5′- GTTGGGAGAGATGAGGTTGG

### Generation of human iPSC-derived microglia-like cells

The microglia differentiation method used in this study has been previously described by us in detail[10]. In summary, iPSC colonies were maintained in mTeSR Plus media (Stem Cell Technology, 100-0276) and passaged every 4–6 days with 0.5 mM ethylenediaminetetraacetic acid (EDTA, Invitrogen 15575020) in Dulbecco's Phosphate-Buffered Saline (DPBS; Corning, 21-031-CV). Six-well plates coated for at least 2 h with 10 μg/mL of Cell Adhere Laminin 521 (Stem Cell Technology, 200-0117) diluted in DPBS containing calcium and magnesium (Corning, 21-030-CV) were used to increase the yield of downstream progenitors as described[64]. Before differentiation, expression of the pluripotency markers octamer-binding transcription factor 4 (OCT4) and SOX2 were confirmed in all iPSC lines by immunofluorescence staining (Supplementary Fig. 1). Induced PSCs were differentiated within five passages into microglia-like cells (iMGs).

On day 0, iPSCs were differentiated into embryoid bodies (EBs)[64,65]. When iPSC colonies were 80% confluent, cells were dissociated using TrypLE reagent (Life Technologies, 12-605-010) for 2 min at 37 °C, gently detached using a cell lifter, and collected in DPBS. The cells were pelleted by centrifugation at $500 \times g$ for 1 min and resuspended in EB media (Table 1) to induce a mesoderm lineage. Cells were seeded at a density of $10^4$ cells/well into low adherence 96-well plates (Corning 7007) and centrifuged at $125 \times g$ for 3 min. Half the volume of media was replaced with fresh media on day 2.

On day 4, myeloid differentiation was induced as previously described[5,6] with the following exceptions. EBs were plated in 6-well plates pre-coated for at least 1 h with Matrigel (Corning, CB-40230), and EBs were diluted in KnockOut Dulbecco modified Eagles media (DMEM)/F12 (Gibco, 12660012) to improve their attachment onto the plate[64]. EBs were maintained with PMP media (Table 1) and a full media change was performed every 5–7 days. After 28 days, primitive macrophage precursors (PMPs) were harvested weekly for up to 3 months

**Table 1 | Media and reagents used to generate iPSC-derived microglia-like cells**

| Reagent | Final concentration | Supplier | Catalog No. |
|---|---|---|---|
| **EB media** | | | |
| mTeSR PLUS | 1× | Stem Cell Technologies | 100-0276 |
| ROCK inhibitor Y27632 | 10 μM | Fisher Scientific | BD 562822 |
| BMP-4 | 50 ng/ml | Fisher Scientific | PHC9534 |
| VEGF | 50 ng/ml | PeproTech | 100-20 A |
| SCF | 20 ng/ml | PeproTech | 300-07 |
| **PMP media** | | | |
| X-VIVO 15 | 1× | Lonza | 12001-988 |
| GlutaMAX | 1× | Gibco | 35050061 |
| Penicillin/Streptomycin | 100 U/mL | Gibco | 15140122 |
| 2-mercaptoethanol | 55 μM | Gibco | 21985023 |
| M-CSF | 100 ng/ml | PeproTech | 300-25 |
| IL-3 | 25 ng/ml | PeproTech | 200-03 |
| **iMG media** | | | |
| Advanced DMEM/F12 | 1× | Gibco | 12634010 |
| GlutaMAX | 1× | Gibco | 35050061 |
| N2 | 1× | Gibco | 17502-048 |
| Penicillin/Streptomycin | 100 U/mL | Gibco | 15140122 |
| 2-mercaptoethanol | 55 μM | Gibco | 21985023 |
| IL-34 | 100 ng/ml | PeproTech or Biologend | 200-34 or 577904 |
| M-CSF | 5 ng/ml | PeproTech | 300-25 |
| TGF-β | 50 ng/ml | PeproTech | 100-21 |

by completely removing the PMP media (containing the cells) and then replacing fresh PMP media back into the well. Harvested cells were plated on Primaria-treated cell culture plates (Corning, 353846, 353847, 353872) at a density of ~$10^5$/cm$^2$ and terminally differentiated into iMGs for 10–12 days by addition of iMG media (Table 1) that contains 100 ng/ml IL−34[5], 5 ng/ml M-CSF[66], and 50 ng/ml TGF-β[67]. Half the volume of media was replaced with fresh media every 3–4 days. All iMG analyses were performed at 10–12 days of microglia differentiation.

## Immunostaining, imaging acquisition, and quantification of iMGs

Immunostaining for fluorescence microscopy was performed as previously described[27]. Briefly, cells grown on glass coverslips (Carolina Biological Supply, 633029) were fixed with 4% paraformaldehyde (Fisher Scientific, AAA1131336) for 15 min and permeabilized with 1% Triton X-100 (Sigma, T9284) for 10 min. Cells were blocked with 50 mM ammonium chloride (Cambridge Isotope Laboratories, NLM-467-5), 1% bovine serum albumin (BSA; Fisher BioReagents, BP1600), 2% goat serum (Sigma−Aldrich G9023), 0.01% Triton X-100 in PBS for 1 h and probed with the antibodies listed in Data S6 for 1 h. Cells stained for TBC1D15, TOMM20, p62, and LC3 were instead blocked and permeabilized with 0.1% saponin in 1% BSA/PBS for 15 min. After 3–5 washes, cells were incubated with secondary antibodies at 1:2000 for 1h[68]. For samples in which F-actin levels were analyzed, Alexa Fluor 647-conjugated Phalloidin (Invitrogen, A22287) was applied for 45 min following the manufacturer's recommendations. Cells were stained with DAPI (Sigma−Aldrich, D9542) and coverslips were mounted with ProLong Gold anti-fade reagent (Invitrogen, P36930).

Images were acquired with a 40× air objective, a 63×-oil or a 100× oil-immersion objective on a Leica DMI 6000B inverted fluorescent microscope equipped with a Leica DFC365 FX camera

and using AF6000 Leica Software v3.1.0 (Leica Microsystems). Z-stacks were collected with a step size of 0.2 μm. Images of cells stained with antibodies other than those targeting p62 and LC3 were deconvolved using the LAS AF One Software Blind algorithm (10 iterations) from Leica Software v3.1.0. Brightness and contrast were equally adjusted post-acquisition across all images using ImageJ (Fiji, 2.3.051, http://imagej.nih.gov/ij/index.html, USA) to improve target visualization. Images are presented as maximum projections unless otherwise noted.

To calculate the relative area or intensity level of a target protein, individual cells were automatically thresholded using the "Default" method in ImageJ. The "Measure" function was applied to obtain the area and the mean intensity from 60–75 cells across at least three independent differentiations. The experimentalist was blinded to the genotype of the cells. The mean for mutant iMGs was normalized to that of WT iMG corresponding to the same independent differentiation.

For the colocalization analyses of TBC1D15, PFN1, and TOMM20 signals, the JaCoP plugin (Fiji, ImageJ) was used to obtain the Pearson's correlation coefficient. Five fields of view (FOV) were analyzed per condition from five independent iMG differentiations[69].

For the analysis of LAMP1 subcellular localization, the number of cells that exhibited LAMP1 signal clustered in the perinuclear area as well as the total number of cells were manually counted for 6–7 FOV per condition using the Cell Counter plugin (Fiji, ImageJ). The experimentalist was blinded to the genotype of the cells. The sum of the number of cells with LAMP1 perinuclear localization was divided to the total number of cells in all the analyzed FOV per condition and multiplied by 100 to obtain the percentage. Images from five WT microglia differentiations derived from two independent iPSC lines, five ALS-PFN1 differentiations derived from M114T$^{+/-}$ iPSCs ($n = 3$) and C71G$^{+/-}$ iPSCs ($n = 2$), and three differentiations derived from M114T$^{+/+}$ iPSCs were used in this analysis. A total of 100–200 iMGs were analyzed per group.

For the analysis of p62 and LC3 puncta, individual cells were automatically thresholded using the "Intermodes" method in ImageJ. The "Analyze Particles" function was used to identify puncta >0.05 μm$^2$ and record the number of puncta per cell as well as their individual areas. $n = 147$ WT and 144 C71G$^{+/-}$ iMGs were analyzed across four independent differentiations. From these cells, $n = 960$ WT and 1476 C71G$^{+/-}$ LC3 and $n = 1438$ WT and 2224 C71G$^{+/-}$ p62 puncta were analyzed. Colocalization analyses were performed using the JaCoP plugin to calculate the Pearson's correlation coefficient on 144 WT and 143 C71G$^{+/-}$ cells from the same set, excluding the cells that had no detected p62 puncta.

## Gene expression analysis by real-time quantitative PCR (qPCR)

Total RNA was isolated using Trizol (Invitrogen,15596026) and treated with DNase I (Invitrogen, 18068015) following the manufacturer's protocols. RNA purity and concentration were determined from absorbance (A) measurements using a Nanodrop device (ND-1000). Samples with A260nm/A280nm between 1.8 and 2.2 were considered pure and used for downstream applications. Complementary DNA (cDNA) was synthesized using iScript Reverse Transcription Supermix (Bio-Rad, 9170-8840) according to the manufacturer's instructions, and qPCR was performed using 5× iTaq Universal SYBR Green Supermix (Bio-Rad, 172-5120) with the primer sets listed below in Table 2. The qPCR experiments were conducted at 95 °C for 2 min, 40 cycles at 95 °C for 5 s, and 60 °C for 30 s, with a melt-curve from 65 to 95 °C (in 0.5 °C increments per 5 s) using CFX384 Touch Real-Time PCR Detection System (Bio-Rad, 1845384). Quantification of mRNA was determined using the comparative cycles to threshold CT (ΔΔCT) method, normalizing to *GAPDH* as internal standard and then normalizing to values obtained from WT iMGs for each independent differentiation. Samples were run in triplicate from three independent

**Table 2 | Primer sequences for qPCR**

| Gene | Primers (5'–3') | Reference |
|---|---|---|
| GPR34 | F: CTGGTTGGGAACATAATCGCC<br>R: GGCAGAAGATGAGTAGGAGGTC | OriGene |
| PROS1 | F: AAGAAGCCAGGGAGGTCTTTG<br>R: ACGTGCAGCAGTGAATAACC | 5 |
| P2RY12 | F: GTCATCTGGGCATTCATGTTCT<br>R: ACCTACACCCCTCGTTCTTAC | 93 PrimerBank ID<br>29029604c3 |
| MERTK | F: CTTCTCCATGGCCACAGGTT<br>R: ATACTGAAAAGGTGGGGCGG | 5 |
| SPI-1 | F: CAGCTCTACCGCCACATGGA<br>R: TAGGAGACCTGGTGGCCAAGA | 94 |
| SOX2 | Proprietary sequence | Bio-Rad (qHsaCED0036871) |
| TBC1D15 | F: TGGAAAGACCAATGACCAAGAC<br>R: TCCACTATTACTTCGGCATCCT | 95 |

differentiations or iPSC passages. Averages of technical triplicates per experiment are reported.

### Transcriptome analysis by RNA-sequencing (RNAseq)

Seven independent differentiations from WT iMGs (two different iPSC lines), four independent differentiations from C71G[+/−] iMGs, and three independent differentiations from M114T[+/−] and M114T[+/+] iMGs were lysed with Trizol and total RNA was isolated with Direct-zol RNA Microprep Kit (Zymo Research, R2060) according to the manufacturer's instructions. RNA library preparation and sequencing was conducted by Novogene Co., LTD (Beijing, China) with a read length of 150 base pairs.

Sample sequences were checked for overall quality as well as possible adapter contamination using FASTQC and multiQC tools[70]. FASTQ files were mapped to GRCh38 genome using STAR aligner (2.7.0a) in two-pass mode with a splice aware option[71]. In particular, the options, -outSAMtype BAM SortedByCoordinate was used to produce sorted bam and −sdjbOverhang 100 for optimal splice junction overhang length. Read counts were computed for each transcript based on GENCODE version 33 annotation using HTSeq tool in strand-specific mode[72]. Transcripts with low-count reads were filtered from the dataset by retaining only those with a sum of at least 20 reads in all samples.

Principal component analysis (PCA) was performed including previously reported data sets downloaded from Gene Expression Omnibus with accession numbers GSE110952 from Brownjohn et al. [5] and GSE89189 from Abud et al. [4]. All data sets were batch-effect corrected using ComBat-seq[73] and normalized using the mean of ratios method in DESeq2[74]. The EdgeR library[75] was used for expression-based filtering using the filterbyExpr function and counts per million (CPM) were calculated. PCA was performed with CPM values using the prcomp function and the plotly package for 3D visualization[76] in R.

Differential gene expression analysis was performed using read counts with the DESeq2 package. The Wald test was used to obtain $P$-adjusted values and log2 (fold change) between the WT group ($n = 7$ WT samples from two iPSC lines) and the ALS group ($n = 4$ C71G[+/−] and $n = 3$ M114T[+/−] samples)[74]. Genes with a $P$-adjusted value < 0.05 were considered differentially expressed.

### Lipopolysaccharide (LPS) stimulation and cytokine analysis

Microglia-like cells were treated for 6 and 24 h with 100 ng/ml of LPS from *Escherichia coli* (*E. coli*) O55:B5 (Sigma L2880) that was diluted first in DPBS and then added to the iMG media. After each time point, the supernatant was collected, centrifuged at $14,000 \times g$ for 10 min, and stored at −80 °C. Samples were thawed and the levels of the cytokines were detected with the Human ELISA MAX™ Deluxe Set

containing IL-6 (430504), IL-10 (430604), CCL5 (440804), and TNF-α (430204) from BioLegend according to the manufacturer's protocol. Each condition was measured in duplicate and the average of at least three independent differentiations is reported herein.

### Quantitative proteomics

One million cells from three independent differentiations for WT and C71G[+/−] iMGs were lysed with RIPA buffer (Boston BioProducts, BP-115-500) supplemented with protease inhibitors (Roche, 11836170001) on ice for 15 min. Lysates were cleared through centrifugation (14,000 g for 10 min at 4 °C) and total protein content was quantified with a Bicinchoninic acid assay (BCA) Assay (Thermo Scientific Pierce, 23227) according to the manufacturer's instructions. For each sample, 50 μg of lysate was diluted to a final concentration of 1 μg/μL with 100 mM triethyl ammonium bicarbonate for tandem mass tag (TMT) labeling using an amine-reactive TMT Isobaric Mass Tagging 6-plex Kit (Thermo Fischer, 90061) as previously described[77]. LC–MS/MS experiments were performed in triplicate using an Orbitrap Fusion Lumos Tribid mass spectrometer in OTOT (orbitrap/orbitrap) ddMS2 mode (Thermo Scientific) equipped with a nanoACQUITY Ultra-Performance LC (UPLC; Waters, Milford, MA, USA). The peptides were loaded and trapped for 4 min with 5% acetonitrile (0.1% formic acid) at 4.0 μL/min onto a 100 μm I.D. fused-silica precolumn (kasil frit) packed with 2 cm of 5 μm (200 Å) Magic C18AQ (Bruker-Michrom, Auburn, CA). Next, the peptides were eluted and separated at 300 nL/min by an in-house made 75 μm I.D fused-silica analytical column (gravity-pulled tip) packed with 25 cm of 3 μm (100 Å) Magic C18AQ (Bruker-Michrom, Auburn, CA). The UPLC method used for the separation and elution was a 145 min gradient starting from 10% of acetonitrile (0.1% formic acid) and 90% of water (0.1% formic acid). The MS data acquisition was performed in positive electrospray ionization mode (ESI+), within the mass range of 375–1500 Da with the orbitrap resolution of 120000 with a maximum injection time of 50 milliseconds. Data Dependent Acquisition (ddMS2) was carried out with a 1.2 Da isolation window, with a resolution of 30000 at m/z 200. The maximum injection time for the ddMS2 was set to 110 milliseconds with the customed AGC target with a 38% of HCD collision energy.

Raw data files were peak processed with Proteome Discoverer 2.1.1.21 (ThermoFisher Scientific Inc.) using a Mascot Search Engine (Matrix Science Ltd, Server version 2.6.2) against the Human (Swissprot, V42) FASTA file (downloaded 04/09/2019). Search parameters included Trypsin enzyme, and variable modifications of oxidized methionine (+16 on M), pyroglutamic acid for glutamine (−17 on peptide N-Terminal) and N-terminal acetylation (+42). Fixed modifications were set for the carbamido-methylation on cysteine (+57 on C) and TMT 6-plex (+229 on peptide N-Terminal and Lysine). Assignments were made using a 10-ppm mass tolerance for the precursor and 0.05 Da mass tolerance for the fragments. The FDR (1%) analysis was carried out by Scaffold (version 4.4.4, Proteome Software, Inc.) and Q+ quantitative analysis was carried out for fold change analysis. PFN1 C71G[+/−] and WT were compared with a T-test followed by a Benjamini-Hochberg test for multiple-test correction. Proteins with a $P$-value < 0.00160 were considered significantly different between genotypes. Mass spectrometry data are available via ProteomeXchange with identifier PXD038943[78,79].

### Functional enrichment analysis

Pathway and process enrichment analysis was conducted with differentially expressed proteins ($p$-value < 0.00160) using Enrichr (https://maayanlab.cloud/Enrichr/) with the libraries: Bioplanet 2019, KEGG 2021 Human and GO Cellular Component 2023[80]. All proteins identified by TMT proteomics were used as background and the default settings were selected to perform the analyses. Functional terms with a $p$-value < 0.05 were compiled[81]. Additionally, pathway enrichment was

performed using Metascape[81]. (https://metascape.org) with the following ontology sources: KEGG Pathway, GO Biological Processes, Reactome Gene Sets, Canonical Pathways, Cell Type Signatures, CORUM, TRRUST, DisGeNET, PaGenBase, Transcription Factor Targets, WikiPathways, PANTHER Pathway, and COVID.

## Western blot analyses of cultured cells

Cell lysates were prepared as described under "Quantitative proteomics", except for PFN1 knockdown and lipid droplet analysis in HMC3 cells in which the following lysis buffer was used in place of RIPA: 20 mM Tris HCl, 150 mM NaCl, 1% CA630 with pH 8. Western blotting was performed as previously described[63] except that for the analysis of the autophagy-related proteins, microtubule-associated protein 1 A/1B-light chain 3 (LC3) and sequestosome-1 (SQSTM1/p62), the transfer was carried using a Trans-Blot Turbo Transfer System (Bio-Rad, 1704150) set with the high molecular weight protocol (1.3 A, 25 V, 10 min), with the Trans-Blot Turbo RTA Mini Transfer Kit (Bio-Rad, 1704272) and 0.45 μm Immobilon-FL transfer membranes (Fisher Scientific, IPFL00010). Primary antibodies were used as follows: 1:1000 for rabbit anti-PFN1 (Sigma–Aldrich, P7749), rabbit anti-CTSD (Cell Signaling Technology, 2284 S), rabbit anti-RAB7 (Cell Signaling Technology, 9367 T), rabbit anti-LC3 (Sigma–Aldrich, L7543), guinea pig anti-p62 (PROGEN, GP62-C), mouse anti-VDAC1 (Abcam, ab14734), rabbit anti-RASD2 (Abclonal, A17367); 1:500 for rabbit anti-TBC1D15 (Sigma–Aldrich, HPA013388), and 1:10000 for mouse anti-GAPDH (Sigma–Aldrich, G8795) or rabbit anti-GAPDH (Sigma–Aldrich, G9545). Membranes were incubated with LI-COR secondary antibodies (Li-Cor Biosciences) at 1:10,000 for 1 h as described[68]. Western blots were visualized using the Li-Cor Odyssey system (ODY-2215) and densitometric analysis was performed using Image Studio (Li-Cor, v3.1.4). Data were obtained from at least three independent iMG differentiations or individual HMC3 experiments.

## BODIPY staining and quantification of lipid droplets in iMGs

PMPs were seeded on coverslips and differentiated into iMGs as described above. After 10–12 days of differentiation, half of the media volume was removed and replaced with a staining solution containing BODIPY 493/503 (Invitrogen, D3922) in DMSO, which was further diluted in iMG media (Table 1) for a final concentration of 2 μM BODIPY 493/503[82]. After 15 min at 37 °C, cells were washed twice with DPBS and fixed with 4% paraformaldehyde for 15 min at ambient temperature. Immunofluorescence staining, imaging acquisition, and deconvolution were performed as mentioned in the section Immunohistochemistry, imaging acquisition, and analysis.

Individual cells from maximum projected images were automatically thresholded by using the "Default" method in ImageJ and the total area of lipid droplets (BODIPY fluorescent signal) was measured using the Particle Analyzer plugin (Fiji, Image). The total area from all cells within a condition (i.e., genotype) were averaged per independent differentiation. For each independent differentiation, data from mutant iMGs were normalized to the corresponding WT iMG condition. A total of 60 cells were analyzed per condition across three independent differentiations.

## HMC3 culture experiments and analyses

The Human Microglial Clone 3 (HMC3) cell line (CRL-3304) was obtained from ATCC (American Type Culture Collection). HMC3s were maintained in Eagle's Minimum Essential Media (Corning, 10-009-CV) supplemented with 10% (vol/vol) FBS (Sigma–Aldrich, F4135) under standard culture conditions (37 °C, 5% $CO_2$/95% air). Media was changed every 2–3 days. Cells were split with 0.25% trypsin (Gibco, 15090046) when they reached 80-90% confluence and sub-cultured for further passages. For PFN1 knockdown, third-generation lentiviruses carrying a CSCGW2 lentivector plasmid containing miRNA targeting PFN1 (GCAATAAGGGGTATGGGGTA) or a scramble sequence (TAATCGTATTTGTCAATCAT) under the U6 promoter were produced by VectorBuilder. Cells were transduced at a multiplicity of infection (MOI) of 5 in the presence of 5 μg/mL polybrene. After incubation for 18 h, lentiviral particles were removed by changing the media. Cells were maintained for another five days before processing. For lysis and western blotting, see "Western blot analyses of cultured cells" section above.

For immunofluorescence analyses of HMC3 cells, cells were grown on glass coverslips (Carolina Biological Supply, 633029) and fixed with 4% paraformaldehyde for 15 min. Cells were blocked and permeabilized with 0.1% saponin in 1% BSA/PBS for 15 min at ambient temperature. Cells were then probed for 1 h at ambient temperature with the following antibodies: mouse anti-PFN1 at 1:350 (Genetex, GTX83903) and guinea pig anti-PLIN2 at 1:200 (Fitzgerald Industries, 20R-AP002) listed in Data S6. After 3–5 washes, cells were incubated with secondary antibodies at 1:2000 for 1 h. Cells were stained with DAPI and coverslips were mounted with ProLong Gold anti-fade reagent.

Confocal images were acquired as z-stacks with a step size of 0.4 μm using a 63× oil objective on a Leica TCS SP5II confocal system. Between 5–6 stacks were collected per condition for five independent experiments. Z-stacks were max-projected and the brightness and contrast were equally adjusted post-acquisition across all images using ImageJ. To calculate the relative area of lipid droplets, images were subject to the "Subtract Background" function in ImageJ to improve target visualization and then thresholded using the "Default" method in ImageJ. The "Analyze Particles" function was used to measure the total area occupied by puncta >0.1 μm², which was then divided by the number of cells within the image. The mean area for cells receiving the PFN1-targeting miRNA sequence was normalized to that of the cells receiving the scramble sequence corresponding to the same independent experiment.

## iPSC-derived lower motor neuron (i³ LMN) differentiation

A WTC11 iPSC line with stable insertion of the hNIL inducible transcription factor cassette containing neurogenin-2 (NGN2), islet-1 (ISL1), and LIM homeobox 3 (LHX3) into the CLYBL safe harbor locus was generously provided by Dr. Michael Ward and differentiated as previously described[83]. Differentiation was induced with DMEM/F12 (Gibco, 11330032) supplemented with N2 supplement, GlutaMAX, nonessential amino acids (NEAA; Gibco, 11140050), 10 μM ROCK inhibitor (Fisher Scientific BD, 562822), 0.2 μM compound E (Calbiochem, 565790), and 2 μg/mL doxycycline (Sigma, D9891). After 2 days, the cells were dissociated with Accutase (Corning, 25058CI) and plated on 10-cm cell culture dishes (Corning, 430591) coated with poly-L-ornithine (Sigma, P3655), poly-D-lysine (Sigma, P7405), and 15 μg/mL laminin (Gibco, 23017015) at a density of ~50,000 cells per cm². The cells were maintained with DMEM/F12 containing 10 μM ROCK inhibitor, 0.2 μM compound E, 2 μg/mL doxycycline, 40 μM BrdU, and 1 μg/mL laminin for another 2 days. For motor neuron maintenance, the neurons were cultured with the B-27 Plus Neuronal Culture System (Gibco, A3653401) supplemented with N2, NEAA, GlutaMAX, and 1 μg/mL laminin. Half-media changes were performed every 3–4 days. The cultures were maintained for 28 days before synaptosome purification as described below under "Mouse and human synaptosome purification and labeling".

## Mouse and human synaptosome purification and labeling

Mouse synaptosomes were purified from wild-type C57BL/6J mice postnatal day 18–21. The following steps were performed on ice and under sterile conditions. Cortices were dissected and homogenized in sucrose-HEPES buffer [0.32 M sucrose (Fisher Scientific, S5-500), 5mM N-2-hydroxyethyl piperazine-N-2-ethane sulfonic acid (HEPES; Gibco, 15630080), and complete, mini protease inhibitor cocktail tablets (Roche, 11836153001)] using a glass Dounce homogenizer with

50 strokes. The homogenate was centrifuged at $1200 \times g$ at 4 °C for 10 min to clear cell debris. The supernatant was collected and centrifuged at $15,000 \times g$ for 15 min. The resulting pellet was resuspended in 1 mL of sucrose-HEPES buffer. A discontinuous gradient containing 0.8 M, 1 M, and 1.2 M sucrose (from top to bottom) diluted each in 5 mM HEPES was prepared in 13.2 mL open-top, thin-wall ultraclear tubes (Beckman Coulter, 344059). The collected supernatant was layered over the sucrose gradient and centrifuged at $150,000 \times g$ for 2 h in a swinging bucket rotor at 4 °C. Fractions at the interface between the 1.0 and 1.2 M sucrose layers were collected, diluted to 0.32 M sucrose by adding 2.5 volumes of 5 mM HEPES (pH 7.4), and centrifuged at $15,000 \times g$ for 30 min. The final pellet comprised of synaptosomes was diluted in DPBS and stored at −80 °C until further use.

Human synaptosomes were purified from i³ LMN after 28 days in culture as described before[10,84]. Briefly, i³ LMNs were washed twice with DPBS. Then, 2 mL Syn-Per reagent (ThermoFisher Scientific, 87793) was added to each 10-cm cell culture dish. Neurons were scraped, collected, and centrifuged at $1200 \times g$ for 10 min at 4 °C. The supernatant was collected and centrifuged at $15,000 \times g$ for 20 min 4 °C. The resulting pellet comprised of synaptosomes was resuspended in DPBS and stored at −80 °C until further use.

Mouse and human synaptosomes were labeled with the pH-sensitive dye, pHrodo Red, succinimidyl ester (ThermoFisher Scientific, P36600), or the pH-insensitive dye, Alexa Fluor 546 NHS Ester (AF546; Invitrogen, A20002) in 100 mM sodium bicarbonate solution (pH 8.5; Sigma, S6297-250G) following the manufacturer's instructions. Synaptosomes were stored in 5% dimethyl sulfoxide (DMSO; Sigma, D2650) in DPBS at −80 °C until further use.

### Live-cell phagocytosis assays and drug treatments

The live-cell phagocytosis assay was performed as described previously in detail[10]. Briefly, PMPs were plated in Primaria 96-well plates (Corning, 353872) and allowed to differentiate into iMGs for 10 days as described[10]. Differentiated iMGs were incubated at 10 °C for 10 min and then sonicated pHrodo red-labeled mouse or human synaptosomes were added to the cells. For wells containing iMGs treated with compound, iMGs were pre-treated with 10 μM cytochalasin D (Sigma, C8273) or 100 nM bafilomycin A1 (Med Chem Express, HY-100558) for 30 min prior to administration of synaptosomes, at which point the drugs were diluted to 5uM and 50 nM, respectively. For experiments with rapamycin, iMGs were incubated with 100 nM rapamycin (Combi-Blocks QA-9258) diluted in DMSO for 24 h before the assay was conducted; rapamycin was diluted to 50 nM upon addition of synaptosomes. iMGs were also treated with 0.05% DMSO as vehicle controls. The plate was centrifuged at $270 \times g$ for 3 min at 10 °C to facilitate contact between synaptosomes and iMGs[85]. At least 2 h before the assay was conducted, cell nuclei were stained with NucBlue Live Ready reagent (Invitrogen, R37605) as per the manufacturer's instructions. For phagocytosis of aggregated protein, PFN1 C71G recombinant protein that was described before[29] was induced to aggregate by shaking at 1000 rpm overnight at ambient temperature. The formation of amorphous aggregates was confirmed with transmitted light microscopy using the Leica DMI 6000B inverted fluorescent microscope described above. The aggregated protein was labeled with pHrodo Red and administered to iMGs as described for synaptosomes.

Sixteen phase, blue, and red fluorescence images were acquired per well at 20× every hour for 12 h using the Cytation 5 cell imaging reader (Biotek). Imaging analysis was performed with Gen5 software (Biotek). For each time point, the area of pHrodo signal was calculated and normalized to the cell number obtained by counting the nuclei in the blue channel. The normalized area for three technical replicates was averaged. For each independent differentiation, the phagocytosis index was calculated with the following equation: normalized area at each time point/ (normalized area at 12 h for WT iMGs - normalized

area at 0 h for WT iMGs). The area under the curve was determined using GraphPad Prism v9.3.1. Data from at least three independent differentiations are reported.

### Washout engulfment assay and analysis

iMGs were incubated at 10 °C for 10 min and fed with sonicated AF546- ("0 h time point") or pHrodo red-labeled ("48 h post-washout time point") human synaptosomes. Samples were centrifuged at $270 \times g$ for 3 min at 10 °C and then incubated for 15 min at 37 °C to allow for initial binding and uptake. Unbound synaptosomes were washed away with DPBS and cells were immediately fixed (0 h time point) or incubated at 37 °C for an additional 48 h with fresh iMG media Table 1 to allow time for degradation of engulfed material. Immunostaining and fluorescence imaging acquisition was performed as described above.

At the 0 h time point, the relative engulfment index was obtained by analyzing individual cells from four independent differentiations. Z-stacks were separated by channel and the background was subtracted from the synaptosome channel by using the "Background subtraction" function with a rolling ball radius of 10 pixels in ImageJ. Since AF546-labeled synaptosomes were used for this time point, it was necessary to exclude unbound synaptosomes that were not washed away from the analysis. To this end, three-dimensional (3D) reconstructions of an overlay of Z-stack images for the IBA1 and synaptosome channels was performed using the 3D Viewer plugin (Fiji, ImageJ). A 3D visualization of the cells allowed for identification of synaptosomes that are located inside the cell volume or in direct contact with the cell boundary defined by IBA1 staining. Synaptosomes that did not meet these criteria were manually deleted from the original Z-stack by an experimentalist blinded to the genotype of the cells. After free synaptosomes were removed, Z-stacks were maximum projected and automatically thresholded using the "Otsu" method for the synaptosome channel and the "Mean" method for the IBA1 channel. The total area of the synaptosomes, as well as the total cell area, were measured by using the "Measure" function in ImageJ. The area of synaptosomes per cell was then normalized to the cell area values for at least 25 cells per condition. Data from the mutant line was normalized to the WT line according to each differentiation.

For the 48 h post-washout time point, the volume of pHrodo-labeled synaptosomes located within CD68-positive endo-lysosomal compartments was measured from individual cells across three independent differentiations. The background signal from the synaptosome (red) channel was subtracted using the "Background subtraction" function described above. Background subtraction was also performed for the CD68 (green) channel, except using a rolling ball radius of 50 pixels. To confirm that the synaptosome signal colocalized with CD68 signal, Z-stacks were maximum projected and both channels were superimposed to visually confirm the presence of yellow (i.e., green overlaid with red) structures. Most of the synaptosome fluorescent signal show spatial overlap with CD68. In rare instances when the signal did not colocalize, the synaptosomes were manually removed from the Z-stacks by an experimentalist blinded to the genotype of the cells. Processed Z-stacks were automatically thresholded using the "Otsu" method and the volume of synaptosomes was measured using the macro "Measure Volume of Thresholded Pixels in an Image Stack" (https://visikol.com/blog/2018/11/29/blog-post-loading-and-measurement-of-volumes-in−3d-confocal-image-stacks-with-imagej/) in ImageJ. The values from at least 10 cells per condition were averaged and normalized to the values of WT iMGs for the corresponding differentiation.

The Cytation 5 cell imaging reader was used to acquire live-cell images of iMGs 48 h post-synaptosome washout for quantification of the percentage of pHrodo-positive cells remaining at that time point. Two technical replicate wells were imaged per genotype. Twenty FOVs per technical replicate at 20× magnification were imaged. The number of pHrodo-positive cells was manually counted using the Cell Counter

plugin (Fiji, ImageJ) by an experimentalist blinded to the genotype of the cells. The total number of cells per FOV was counted similarly using the bright field channel. The sum of the number of cells in all the analyzed FOV was divided to the total number of cells per duplicate and multiplied by 100 to obtain the percentage. The average percentage for technical duplicates was reported. A total of 400 and 600 cells were analyzed per technical replicate across four independent differentiations.

## Flow cytometry synaptosome uptake assay and analysis

iMGs were incubated at 10 °C for 10 min and fed with sonicated AF546-labeled human synaptosomes. The cells were centrifuged at $270 \times g$ for 3 min at 10 °C and then incubated for 15 min at 37 °C to allow for initial synaptosome binding and uptake. For wells containing iMGs treated with bafilomycin A1 (Med Chem Express, HY-100558), iMGs were pre-treated with 100 nM bafilomycin A1 for 30 min at 37 °C. Unbound synaptosomes were washed away with DPBS. iMGs were then dissociated with Accutase (Corning, 25058CI) for 5 min at 37 °C. Cells were collected with DPBS into BSA-coated Eppendorf tubes (5% BSA at RT for 2 h or at 4 °C overnight) and then centrifuged at $300 \times g$ for 5 min. Cells were resuspended in 100 uL of 1% FBS/DPBS and measured for AF546 fluorescence using a MACSQuant® VYB Flow Cytometer. The median fluorescence intensity was calculated using FlowJo™ v10.8.1 Software (BD Life Sciences). Data was obtained from three independent iMG differentiations.

## Simple Western analysis of mouse tissue lysates

Mouse tissue lysates were prepared from the brains and spinal cords of 8-week-old WT and PFN1 C71G[+/−] animals. Western blots were performed on a WES 3.8.21 instrument using Simple Western technology, an automated capillary-based size sorting and immunolabeling system (ProteinSimple, Biotechne). All procedures were performed with manufacturer's reagents according to their manual. A quantity of 5 μL of each sample (0.8 μg/μL) was analyzed for target protein expression. Primary antibodies were used as follows: 1:25 for rabbit anti-PFN1 (Cell Signaling Technology, 3237 S) and 1:5000 for rabbit anti-GAPDH (NovusBio NB300-322). Data were obtained from twelve WT (6 male, 6 female) and twelve PFN1 C71G[+/−] (6 male, 6 female) animals and analyzed using the manufacturer-provided Compass software.

## Mouse behavior assays

Mouse motor function and balance was assessed using a rotarod test (Ugo Basile, Model 7650). Eight PFN1 C71G[+/−] mice ($n = 6$ males, $n = 2$ females) and eight WT control mice ($n = 3$ males, $n = 5$ females) were trained once a week for three weeks prior to testing. During testing, mice were subjected to trials under continuous acceleration from 8 to 15 rotations per minute (rpm). Mice underwent three trials per session in succession and without rest. The latency to fall from the rod was measured for each mouse and reported as the average of the three trials. All testing was carried out blinded to genotype and gender. Mice were between 593 and 628 days old at the time of the test.

The same animals above were subjected to grip strength assessments for all limbs using the Grip Strength Meter (Bioseb, BIO-GS3); rotarod and grip strength were assessed on the same day for all mice. Mice were allowed to grasp the grid with all four limbs and then gently pulled back by hand. Mice were tested three times in each session and the maximum peak force in grams prior to grip release was recorded. All testing was carried out blinded to genotype and gender.

## Immunostaining, imaging acquisition, and quantification of tissue sections from aged mice

Mice of approximately 600 days of age were euthanized and transcardially perfused with PBS and 4% paraformaldehyde. Brains and spinal cords were dissected and post-fixed with 4% paraformaldehyde overnight and then equilibrated in 30% sucrose before being embedded in Optimal Cutting Temperature (O.C.T.) compound. Twelve μm coronal cryo-sections were prepared. Sections were blocked in 10% goat serum, 0.3% Triton X-100 in 0.1 M PB at ambient temperature for 1 h. Tissues were incubated overnight at ambient temperature with the following primary antibodies: rabbit anti-IBA1 at 1:500 (Wako, NC9288364), rat anti-CD68 at 1:200 (Bio-Rad, MCA1957), and mouse anti-GFAP at 1:500 (Sigma–Aldrich, G3893). Sections were washed and incubated with fluorophore-conjugated secondary antibodies (Jackson ImmunoResearch Laboratories) at 1:1000 and counterstained with DAPI prior to mounting with ProLong Gold anti-fade reagent (Invitrogen P36930) for subsequent immunofluorescence analysis.

Images of single planes were acquired with a 20× air objective on a Leica DMI 6000B fluorescent microscope equipped with a Leica DFC365 FX camera and using AF6000 Leica Software v3.1.0 (Leica Microsystems). For brain sections, three images were collected from the motor cortex; for spinal cord sections, three images were collected from the left and right ventral horn of the lumbar section, respectively. Brightness and contrast were equally adjusted post-acquisition across all images using ImageJ (Fiji, 2.3.051, http://imagej.nih.gov/ij/index.html, USA) to improve target visualization. Images were automatically thresholded using the "Otsu" method in the CellProfiler software (Broad Institute, v4.2.5, https://cellprofiler.org). The total fluorescent intensity of IBA1, CD68, and GFAP was measured separately using the "MeasureImageIntensity" function in CellProfiler. The values of all analyzed images per mouse were averaged, normalized to the mean intensity of all WT animals, and then presented as individual data points. All analyses were performed by a subject blinded to the genotype.

## Stereotactic injection of dead neurons into mice, imaging and analysis

Primary neurons were prepared from wild-type *C57BL/6J* mouse embryos harvested at day E18.5 following our published methods[86]. Neurons were cultured for nine days before cell death was induced as described before[39] with the following modifications. Briefly, neurons were gently washed with DPBS and detached from the plate by gently pipetting DPBS at the bottom of the well. Cell death was induced by exposing neurons to UV light (302 nm) for 15 min using a Molecular Imager Gel Documentation System XR+ (Bio-Rad). Dead neurons were maintained on ice for 2–3 h until the time of stereotactic injection into mice. Neurons were resuspended in 1 mL of DPBS followed by the addition of equimolar concentrations of the pH-insensitive dye, Alexa Fluor 546 NHS Ester (AF546), and the pH-sensitive dye, CypHer5E NHS Ester (Cy5E; Cytiva, PA15401) at 37 °C for 30 min. Labeled neurons were then diluted in DPBS, harvested by centrifugation at $200 \times g$ for 8 min at 4 °C, and washed with DPBS to remove residual dye as per the manufacturer's instructions. The number of dead cells was determined using Trypan Blue 0.4% staining solution (AMRESCO Inc, K940).

Dead neurons were resuspended in DPBS at a density of 50,000 neurons per microliter for subsequent stereotactic injection into three-month-old PFN1 C71G[+/−] and WT littermates. Four PFN1 C71G[+/−] mice (2 males, 2 females) and five WT control mice (2 males, 3 females) were included in this study. Mice were initially anesthetized through 3% isoflurane inhalation and anesthesia was maintained with 1.5% isoflurane throughout the procedure. Stereotactic injections were performed according to a previously described protocol with the following modifications[87]. Briefly, 1 uL containing 50,000 dead neurons was injected into the right hemisphere in the motor cortex at the location of 2 mm in front of bregma and 1 mm right from the midline at the depth of 0.5 mm below the brain surface. As a control, 1 uL of saline solution (Bioworld, 40120975-2) was injected contralaterally (2 mm in front of bregma, 1 mm left from the midline, and a 0.5 mm depth below the brain surface). Injections were performed at a speed of 200 nL/min using a stereotaxic instrument (KOPF, Model 900LS). Anesthesia was discontinued after the operation was completed. Buprenorphine

(1 mg/kg, subcutaneously), cefazolin (500 mg/kg, intramuscularly), meloxicam (5 mg/kg, subcutaneously) were administered prior to the end of surgery session. Animals were placed on top of a heating blanket and their recovery was monitored for ~15 min and until the mice were ambulatory. After recovery, animals were returned to their home cages for 72 h, at which time animals were euthanized with an overdose isoflurane and transcardially perfused with DPBS and 4% paraformaldehyde for subsequent tissue processing.

Mouse brains were dissected and post-fixed with 4% paraformaldehyde overnight, placed in 30% sucrose in 0.1 M phosphate buffer (PB), and allowed to sink to the bottom of the conical tube before sectioning. Forty μm coronal sections were prepared, encompassing the entire site of injection as determined by the presence of AF546 fluorescent signal. The following tissue processing steps were conducted with agitation in a plate rocker at ambient temperature. Floating sections were blocked in 10% goat serum, 0.01% Triton X-100 in 0.1 M PB for 1 h. Sections were incubated overnight with rabbit anti-IBA1 antibody at 1:500 (Wako Chemical USA, NC9288364) diluted in blocking buffer at 4 °C. Sections were washed and incubated with Alexa fluor-488 secondary antibody (Jackson ImmunoResearch, 711-545-152) at 1:1000 in blocking buffer for 1 h and counterstained with DAPI prior to mounting with ProLong Gold anti-fade reagent (Invitrogen, P36930) for subsequent immunofluorescence analysis.

Every tissue section that contained AF546 fluorescent signal from the site of dead neuron injection was imaged and analyzed by an experimentalist who was blinded to the genotype of the tissues. Z-stack images with a step size of 0.2 μm were acquired with a 20X air objective on a Leica DMI 6000B inverted fluorescent microscope equipped with a Leica DFC365 FX camera and AF6000 Leica Software v3.1.0. Z-stacks were sum projected and the background was subtracted using ImageJ. Images were automatically thresholded using the "Otsu" method in the AF546 channel (constitutively fluorescent dead neurons) and the Cy5E channel (dead neurons in acidic compartments) independently. The area of the signal in each channel was assessed with the "Measure" function in ImageJ. The area of AF546 signal within all AF546-positive brain slices for each mouse was summed, resulting in a total "AF-neuron" area. The same procedure was carried out for Cy5E signal. The total area of the fluorescent signal in Cy5E channel was divided by the total AF-neuron area to determine the fraction of dead neurons located within acidic compartments per mouse.

To quantify the IBA1 signal intensity, a region of interest containing the injection site was manually drawn and three concentric rings were automatically generated using an ImageJ macro (Supplementary Code 1). The injection site was excluded from analysis. The total signal intensity of each ring was calculated using the "Measure" function (ImageJ) and normalized to the number of IBA1-positive cells counted manually. The values of all the analyzed brain slices per mouse were average and presented as individual data points. All the analyses were performed by a subject blinded to the genotype.

Confocal images shown in Fig. 5 were acquired with a 40× oil objective on a Leica TCS SP5II (S/N: 5100001537) confocal microscope using the LAS AF software. Z-stacks were collected with a step size of 0.5 μm.

### De-quenched-BSA (DQ-BSA) assay
PMPs were plated in Primaria 24-well plates (Corning, 353847) at a density of 100,000 cells per well for microglia differentiation. iMGs were pulse-labeled with 12 μg/mL green DQ-BSA reagent (Invitrogen, D12050) for 1 h, washed with DPBS, and subjected to live-cell imaging after adding fresh iMG media (Table 1). As a negative control, cells were treated with 200 nM of bafilomycin A for 30 min before the assay. Twenty phase and green fluorescence images were acquired per well from two technical replicates at 20× using the Cytation 5 cell imaging reader (Biotek). Images were acquired immediately after washing and every hour for four hours. Images were analyzed to obtain total DQ-

BSA intensity values and the number of cells by counting the cell nuclei from all the acquired fields of view with Gen5 software. Total intensity values were divided normalized to the number of cells and average value of the technical replicates are reported.

### LysoSensor DND-189 staining and analysis
PMPs were plated in Primaria 24-well plates (Corning, 353847) at a density of 100,000 cells per well for microglia differentiation as described above. On day 10, iMGs were loaded with 1 μM LysoSensor Green DND-189 (Invitrogen, L7535), and incubated for 20 min at 37 °C. This pH sensor dye produces higher fluorescent signal in acidic environments allowing for a relative comparison of the acidification state in intracellular compartments. Cells were washed twice with PBS and fresh iMG media (Table 1) was added before live-cell imaging. At least 2 h before the experiment, cell nuclei were stained with NucBlue Live Ready (Invitrogen, R37605) reagent for quantification of the number of cells in each well as per the instructions from the manufacturer. Twenty phase, green and blue fluorescence images were acquired per well from two technical replicates at 20× using the Cytation 5 cell imaging reader (Biotek). Imaging analysis was performed with Gen5 software, where the total green, fluorescent signal intensity was normalized to the cell number. The average of both technical replicates was normalized to the values of WT iMGs for each independent differentiation to obtain the DND-189 relative intensity. Three independent differentiations were used per sample condition. To confirm that the DND-189 relative intensity is dependent on the pH of acidic compartments, lysosomal acidification was inhibited by pre-treating the cells with 200 nM of bafilomycin A for 30 min.

### Differential scanning fluorimetry (DSF) assay with PIPs
Recombinant PFN1 proteins (WT, C71G, and M114T) for DSF experiments were described in a previous study[29]. The 1,2-dioctanoyl-sn-glycero-3-phospho-(1'-myo-inositol-4',5'-bisphosphate) 08:0 (PIP2; Avanti Polar Lipids, 850185 P) and phosphatidylinositol 3-phosphate diC8 (PI3P; Echelon Biosciences, P-3008) were dissolved in PBS to the different concentrations (20 μM, 50 μM, and 200 μM) and mixed with 25X SYPRO Orange (Invitrogen #S6651). Purified WT or mutant PFN1 proteins were added to each sample tube to a final concentration of 20 μM and plated in duplicate in 384-well plates (Bio-Rad, HSR-4805). The samples were subjected to heat denaturation using CFX384 Touch Real-Time PCR Detection System. The temperature gradually increased from 25 °C to 70 °C, with 0.3 °C increments every 5 s. The fluorescence intensity was acquired at each time point with a HEX detector (excitation 515–535 nm, emission 560–580 nm). The fluorescence intensity values from duplicate samples were averaged and plotted as a function of temperature to produce melting curves in GraphPad Prism. Melting curves were fit with a Boltzmann's sigmoidal function in GraphPad Prism to determine the apparent melting temperature ($T_m$). The difference between the $T_m$ of each experimental group and the no-lipid control condition was reported as the $\Delta T_m$. Two to three independent experiments were performed per condition as indicated in the legends.

### Recombinant PFN1 protein expression for NMR experiments
WT and PFN1 M114T variants were cloned into a modified pet28 vector with a SUMO tag between BamHI restriction site. Both WT and M114T variants were expressed in BL21(DE3) *Escherichia coli* competent cells and isotopically labeled with $^{15}N$ by growing the cells in M9 media containing 1 g of $^{15}NH_4Cl$ per liter. The cells were grown at 37 °C to an $OD_{600}$ of 0.8 and then induced using 1 mM isopropyl-β-d-1-thiogalactopyranoside (IPTG; GoldBio, 2481 C) for 8 h at 30 °C. Cells were harvested and lysed by sonication in 40 mL of buffer containing 50 mM Tris HCl pH 8, 300 mM NaCl, 1 mM Tris (2-carboxyethyl) phosphine (TCEP; GoldBio, TCEP25), and 25 mM imidazole (Fischer Scientific, BP-305-50). Lysates were centrifuged at 20,000 rpm at 4 °C

for 1 h and passed through a pre-equilibrated 15 mL prepacked Histrap™ HP nickel column (Cytiva, 17524802), washed with 5 column volumes of 50 mM Tris HCl pH 8, 300 mM NaCl, 1 mM TCEP and 25 mM imidazole, and eluted with 50 mM Tris HCl pH 8, 300 mM NaCl, 1 mM TCEP, and 300 mM imidazole. The SUMO tag was cleaved off with 5 mL Ubiquitin-like protease 1 (ULP1) at a 1:5 ULP1 to protein ratio during an overnight dialysis at 4 °C[88]. The protein is then passed through a second round of nickel column purification. The purified protein was buffer exchanged into 50 mM phosphate buffer pH 6.5, 150 mM NaCl, 200 µM TCEP, 1 mM EDTA, and 50 mM L-Arg and L-Glu by overnight dialysis then concentrated using a 10 kDa Centriprep concentrator (Millipore, UFC901024). The concentrated protein was passed through a pre-equilibrated size exclusion column (Cytiva Custom Superdex ™ 75 Increase HiScale ™ 16/40, CP 19-186).

## NMR binding studies

Binding of PFN1 WT and M114T to PI3P (described above in the DSF experiments) was monitored via NMR spectroscopy. $^{15}N$-$^1H$ heteronuclear single-quantum coherence (HSQC) spectra were collected at 293 K on a Varian Inova spectrometer (Palo Alto, CA) operating at 600 MHz equipped with a triple-resonance cold probe. Data processing was performed using NMRPipe[89] and Sparky software[90]. Uniformly $^{15}N$ labeled PFN1 WT and M114T mutant were dissolved in 600 µL of buffer (2 mM HEPES, 0.2 mM $CaCl_2$, 0.2 mM ATP, 50 mM L-Arg, and 50 mM L-Glu at pH 7, 92% $H_2O$/8% $^2H_2O$) at initial concentrations of 50 µM. PI3P was dissolved in the same buffer to achieve a stock concentration of 2.7 mM. A titration was performed at 20° C by adding increasing amounts of PI3P to reach the following concentrations: 10 µM, 20 µM, 30 µM, 40 µM, 50 µM, 67.5 µM, 85 µM, 100 µM, 125 µM, 150 µM, 175 µM, 202 µM, 250 µM, 350 µM, 500 µM, 750 µM, 1100 µM, 1450 µM for PFN1 WT and 10 µM, 20 µM, 30 µM, 40 µM, 50 µM, 67.5 µM, 85 µM, 100 µM, 125 µM, 150 µM, 200 µM, 250 µM, 350 µM, 500 µM, 750 µM, 1100 µM, 1450 µM for PFN1 M114T. A $^{15}N$-$^1H$ HSQC spectrum was collected after each addition of PI3P. The chemical shift perturbations were calculated for each PI3P titration point according to the following Eq. (1):

$$\Delta = \sqrt{\Delta\delta_H^2 + (0.152\Delta\delta_N)^2} \qquad (1)$$

where $\Delta\delta_H$ and $\Delta\delta_N$ are the $^1H_N$ and $^{15}N$ chemical shift differences measured for each residue in each $^{15}N$-$^1H$ HSQC spectrum.

The apparent $K_d$ was derived from global fitting of the following residues: Ala 5, Tyr 6, Ile 7, Asn 9, Asp 13, Gln 17, Asp 18, Trp 31, Lys 104, Lys 107, Tyr 128, Glu 129, Met 130, Ala 131, Leu 134 for PFN1 WT, and Ala 5, Tyr 6, Ile 7, Asp 13, Gln 17, Asp 18, Ser 56, Ser 57, Glu 129, Met 130 and Leu 134 for PFN1 M114T. The residue-specific titration curves were fitted using the following Eq. (2) to determine $K_d$ using Matlab[91,92],

$$\Delta = \Delta_{max} \frac{L + P + k_d - \sqrt{(L + P + k_d)^2 - 4LP}}{2P} \qquad (2)$$

where $\Delta_{max}$ is the maximum chemical shift perturbation, L and P are the total ligand and protein concentrations and $K_d$ is the apparent dissociation constant.

## Statistical analysis

Statistical analyses were calculated using GraphPad Prism v9.3.1. Statistical tests are stated in the figure legends. The number of independent experiments is stated in the corresponding methods section and in the figure legends.

## Reporting summary

Further information on research design is available in the Nature Portfolio Reporting Summary linked to this article.

## Data availability

RNAseq data was deposited at GEO under the accession number GSE254069. Mass spectrometry data are available via ProteomeXchange with identifier PXD038943. Any other data generated and/or analyzed during the current study are available in the Source Data files. Source data are provided with this paper.

## Code availability

ImageJ analysis script used to measure Iba1 signal in dead neuron injections tissue is in the supplement.

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

## Acknowledgements

We are grateful to Dr. Michael Ward (NINDS, MD) for providing the WTC11 iPSC line with the hNIL cassette, Dr. Jeffery Kelly (the Scripps Research Institute, CA) and his lab for providing Rapamycin and advice, Drs. Yen-Chen Lin (UMass Chan) and Desiree Baron (UMass Chan) for help with iPSC culture, Dr. Anthony Giampetruzzi (UMass Chan) for help with the Cytation 5, Dr. Travis Faust for advice on mouse studies, the UMass Chan Mass spectrometry Facility for the proteomics experiments, and the UMass Chan Transgenic Animal Modeling Core for in vitro fertilization of mutant PFN1 mice, the Cellular Engineering Service at The Jackson Laboratory for expert assistance with gene-editing of iPSCs described in this manuscript. D.A.B. is supported by the NIH/NINDS (R01 NS108769, R21 NS120126), NIH/NIGMS (R01GM137529, R01 GM147677), the Department of Defense (W81XWH202071/PRARP), the Angel Fund for ALS research, the Radala Foundation and the Robert Packard Center for ALS Research. This project has also been supported by the Dan and Diane Riccio Fund for Neuroscience (UMass Chan; to D.A.B. and D.P.S.) and NIH/NIGMS R01GM137529 (F.M. and D.A.B.). We are also grateful for the following support: NIMH-R01MH113743 (D.P.S.); NINDS-R01NS117533 (D.P.S.); NIA-RF1AG068281(D.P.S.); U54OD020351 (to the Jackson Laboratory Center for Precision Genetics, C.L.).

## Author contributions

S.F., J.J. and D.A.B. planned and oversaw all aspects of the study. S.F. and J.J. performed and analyzed most of the experiments. D.H.G performed staining and Western blot analyses. M.M. and F.M. performed the NMR experiments. D.H.G and M.M. contributed equally to this manuscript as co-second authors. J.Z., K.S. and D.C. performed all the animal experiments and contributed to the analyses. S., P.J.K. and J.E.L. performed the RNASeq analyses. M.U. and D.H.G. performed the DSF assays. P.D. and E.T.L. performed the PCA analysis of the RNASeq results, M.S.R. contributed to the TMT proteomics analyses and PFN1 knockdown studies, M.F.M. created the model figure, J.A.M. and W.C.S. created the PFN1 iPSC lines, S.B. provided recombinant PFN1 protein, M.S.-E. generated the PFN1 miRNA, J.A.N. helped with confocal microscopy analyses, C.L. and J.E.L. provided the PFN1 mouse model. D.P.S. provided critical input on the experiments and direction of the manuscript. S.F., J.J. and D.A.B. wrote the manuscript with input and revisions from all authors.

## Competing interests

The authors declare no competing interests.

## Additional information

[1]Department of Neurology, University of Massachusetts Chan Medical School, Worcester, MA 01605, USA. [2]Translational Science Program, Morningside Graduate School of Biomedical Sciences, University of Massachusetts Chan Medical School, Worcester, MA 01605, USA. [3]Neuroscience Program, Morningside Graduate School of Biomedical Sciences, University of Massachusetts Chan Medical School, Worcester, MA 01605, USA. [4]Department of Biochemistry and Molecular Biotechnology, University of Massachusetts Chan Medical School, Worcester, MA 01605, USA. [5]Biochemistry and Molecular Biotechnology Program, Morningside Graduate School of Biomedical Sciences, University of Massachusetts Chan Medical School, Worcester, MA 01605, USA. [6]Department of Neurosurgery, The First Affiliated Hospital of Chongqing Medical University, Chongqing 400016, China. [7]Department of Microbiology and Physiological Systems, University of Massachusetts Chan Medical School, Worcester, MA 01605, USA. [8]Department of Molecular, Cell and Cancer Biology, University of Massachusetts Chan Medical School, Worcester, MA 01605, USA. [9]Department of Genomics and Computational Biology, University of Massachusetts Chan Medical School, Worcester, MA 01605, USA. [10]The Jackson Laboratory for Genomic Medicine, Farmington, CT 06032, USA. [11]Horae Gene Therapy Center, University of Massachusetts Chan Medical School, Worcester, MA 01605, USA. [12]Department of Pediatrics, University of Massachusetts Medical School, Worcester, Worcester, MA 01605, USA. [13]The Jackson Laboratory Center for Precision Genetics, Rare Disease Translational Center, Bar Harbor, ME 04609, USA. [14]Department of Neurobiology, Brudnick Neuropsychiatric Research Institute, University of Massachusetts Chan Medical School, Worcester, MA 01605, USA. [15]These authors contributed equally: Salome Funes, Jonathan Jung. ✉e-mail: Daryl.Bosco@umassmed.edu

