## [Peer Review File · Nature Communications]

Expression of ALS-PFN1 impairs vesicular degradation in iPSC-derived microgliaREVIEWER COMMENTS

Reviewer #1 (Remarks to the Author):

The manuscript by Funes and collaborators is an interesting study on the functional alterations of iPSC-derived microglial cells bearing ALS associated mutations in the PFN1 gene. The authors nicely demonstrate how mutations in this proteins affect the ability of microglial-like cells to perform the phagocytic process, lipid metabolism and autophagy and speculate that these alterations may play a role in ALS disease progression.

However, additional experiments could be considered to strenght the manuscript conclusions (or the conclusions should be modified). In particular, while the data in vitro are convincing of a role for PFN1 in microglial functioning, less strong is the connection with the pathological effects of PFN1 mut-microglial-like cells in neurodegenerative diseases (last three lines of the abstract).

Major:

1) the authors demonstrate that rapamycin reverts some phenotypic alterations of PFN1-mut microglial- like cells, and these results are presented as a demonstration of the role of autophagic process in some steps of lipid metabolism and phagocytosis alterations. It would be interesting to see the effects of rapamycin treatment on PFN1 mut mice models of ALS. Is rapamycin administration able to reduce disease progression in these animals? These information would significantly strenght the data and would demonstrate the author's hypothesis of a role for microglia in PFN1-ALS.

2) In dead neuron injection experiments the authors used both male and female mice, but the results are shown together. Did they observe sex-related differences?

Minor

1) More info about PFN and ALS should be given in the introduction. The connection is only weakly introduced at page 3 line 84, where the authors state that PFN1 mutations are causative for ALS, but they should better explain its incidence among ALS cases and if mutations of this gene give a 100% ALS disease, the severity of this form in comparison with fus, tdp43, sod1 and c9orf72-mut forms.

2) The animal model used must be introduced. Which signs of the disease were present in the PFN1 C71G+/- mice generated with CRISPR/Cas9 technology? When do they appear in the animal life?

Reviewer #2 (Remarks to the Author):

Funes et al. describe in their manuscript the effect profilin-1 (PFN1) mutation in human iPS cells derived microglia. Using CRISPR/Cas9 gene-editing they introduced ALS-linked C71G+/-, M114T+/- and M114T+/+ mutations in human induced pluripotent stem cells and then differentiating them into microglia like cells (iMG), they demonstrate deficit in phagocytosis and lipid metabolism. Further, they showed that enhanced binding of mutant PFN1 to the autophagy signaling molecule PI3P, as an underlying cause of defective phagocytosis. Although overall the work is important for understanding how PFN1 mutation affects microglial function, the study remains highly descriptive and correlative. It lacks, in my view, further mechanistic insights into how PFN1 mutation affects microglial phagocytosis.

Major concern

1. In figure 1. authors assume LPS induced production of cytokines in PFN1 mutation carrying iMG and control is similar using C71G+/- only. However, the data regarding M114T mutation carrying iMG is missing. Could M114T mutation have similar response?

2. In figure 2, authors first showed that C71G+/-, M114T+/-, M114T+/+ mutation cause PFN1 protein decrease by 60, 30 and 80 % respectively. However, lipid accumulation increase observed in C71G+/-, M114T+/+, was not observed in M114T+/- . Is there any correlation between amount of PFN1 decrease and accumulation of lipid droplet in iMG? Also, from the data presented in current manuscript is insufficient to indicate whether the autophagosome dysfunction is due to decreased PFN1 protein or due to abnormal protein binding. So, it advised that, authors should perform knockdown of PFN1 in iMG or HMC3 cell line to show its effect on to lipid accumulation.

3. Authors have shown that C71G+/-, M114T+/-, M114T+/+ mutation in PFN1 increase TBC1D15 in fig. 3. Authors also show that both wild type mutant and PFN1 show interaction with TBC1D15 by immunohistochemistry. However, PFN1 knockdown in human microglial cell line HMC3 leads to decrease in TBC1D15. Although this upregulation of TBC1D15 in PFN1 mutant iMG led authors to further investigate phagocytotic pathway, the explanation and data they have provided to this discrepancy remains unsatisfactory. First of all, the interaction between TBC1D15 and PFN1 remains unknown. Authors can determine this interaction as they have done in figure 8. Next the authors can determine how PFN1 mutation affect its interaction as well as degradation of TBC1D15 to clarify its role in the processing of phagocytotic products. Although authors have stated that the upregulation of TBC1D15 as a compensatory mechanism the defect in the processing of phagocytotic products, it may well be the cause of this dysfunction. This issue needs to be addressed.

4. Authors have assumed that all PFN1 mutation affects iMG in the same way even though their own data contradicts this assumption as shown in figure 2 i. They have generalized that since C71G+/- mutant iMG do not have defect in synaptosome uptake, it must be true for M114T+/-, M114T+/+ mutant iMG. However, the it may not true. So, it is advisable that they include the data for M114T+/-, M114T+/+ mutant iMG in Figure 4. Also, from the graph presented in Figure 4 k, it is hard to appreciate difference as they have shown.

5. Using PFN1 C71G+/- mice authors have shown that higher number of dead neurons in microglial acidic compartments in figure 5. Further they showed that microglia in PFN1 C71G+/- mice that showed increased dead neuron in phagosome and higher expression of Iba1 (figure 5b-e). Is the number of microglia and Iba1 intensity between PFN1 C71G+/- and control mice similar at basal condition? Also, it is hard to appreciate that higher number of dead neurons in microglial acidic compartments of PFN1 C71G+/- microglia from the image presented in fig 5 b, it would be nice if they could present more convincing image to further support their data. Also, what cause this decrease in microglia in ring 3 region in PFN1 C71G+/- mice? Also do microglia in these mice show accumulation of lipid, increase in TBC1D15 and defect in phagosome processing? Additionally, is Iba1 expression in iMG increased when they are presented with dead neurons or synaptosome?

6. Authors have shown that rapamycin attenuate autophagic as well as phagocytotic dysfunction in PFN1 mutant iMG. Do addition of rapamycin normalize EEA1 expression as well as LC3II/LC3I ratio? Also, in addition to graph, authors should include the immunocytochemistry image in the main or supplementary figure to show that rapamycin treatment affect phagocytotic index. The information regarding how is AUC calculated in Fig 7 i, is missing. It should be included in the method section.

7. As mentioned above, authors assumed that all PFN1 mutation affects iMG in the same way, for protein-protein interaction with PI3P, they have used M114T mutation and generalized this data to all mutation affecting PFN1. As mentioned above each mutation may affect PFN1 function in microglia in different way, so the effect of each mutation should be individually assessed. Why do authors use only M114T mutation for protein-protein interaction instead of C71G? Authors should include PFN1 C71G data to show that similar interaction occurs PI3P.

8. Authors should perform co-culture to show that PFN1 mutation in microglia affect neurons under basal and stressed condition to show the to clarify how microglia dysfunction affect neurons in non-cell autonomous way.

Minor points

1. There are two GAPDH band in figure 7 c. Please remove one GAPDH band.
2. Also which mutation do authors mean by ALS-PFN1 in 7 h. Please change it to C71G+/- for consistency so that is easy for readers to understand.
3. The authors have stated that "Area under the curve (AUC) determined from 7 i" in figure 7 i. This mistyping should be rectified.

Reviewer #3 (Remarks to the Author):

In this paper by Funes et al., the consequences of PFN1 mutations in iPSC-derived microglial-like cells are studied. As PFN1 is expressed at higher levels in microglia than neurons, and microglial activation is a hallmark of ALS pathology the study is relevant to the field. The engineered mutant lines (with two different mutations) differentiate with normal efficiency into microglial cells and displayed similar activation in response to immune stimulation. However, they display some changes in proteome/transcriptome and a problem of clearance of vesicles. The experiments are clearly presented, a large quantity of data is presented. Several observations are made including an accumulation of lipid droplets, increased expression of TBC1D15, perinuclear accumulation of perinuclear LAMP1-positive vesicles, reduced autophagic degradation of vesicles and increased PIP3 binding with reduced PFN1 stability. The relation between these different phenotypes is not always clear and makes the paper heavy to digest.

Specific comments:

- Figure 1: the microglial markers and the responsiveness of the iMGs are only shown for the C71G+/- mutants, but not for the M114T +/- and +/+ lines. It would be of interest to add this as well as it provides extra gene-dose information.
- a proteome and transcriptome analysis have been performed. The results have been analyzed and presented separately in the paper. A combined analysis would be of interest. Where some of the changes at protein level visible at transcript level as well? and vice versa?
- Figure 3: based on the staining experiments, the authors conclude that there is colocalization of TBC1D15 and PFN1. In the images only a partial colocalization is visible. What could be the importance of this finding? How is it linked to what is presented in later figures in the paper?
- Figure 4: Bafilomycin abolishes the pHrodo signal as expected. Was there an effect on the engulfment index and on the volume occupied by pHrodo-synaptosome signal?
- Figure 7: was there still an increase in LC3-II and p62 upon bafilomycin treatment?

REVIEWER COMMENTS

Reviewer #1

Reviewer #1, comment #1 (R1.1): The manuscript by Funes and collaborators is an interesting study on the functional alterations of iPSC-derived microglial cells bearing ALS associated mutations in the PFN1 gene. The authors nicely demonstrate how mutations in this protein affect the ability of microglial-like cells to perform the phagocytic process, lipid metabolism and autophagy and speculate that these alterations may play a role in ALS disease progression.

However, additional experiments could be considered to strengthen the manuscript conclusions (or the conclusions should be modified). In particular, while the data in vitro are convincing of a role for PFN1 in microglial functioning, less strong is the connection with the pathological effects of PFN1 mut-microglial-like cells in neurodegenerative diseases (last three lines of the abstract).

Response R1.1: We thank the reviewer for their positive and constructive feedback on our study. We take the Reviewer's point and have modified the text in the Abstract and Introduction accordingly. Additionally, while this manuscript was under submission/revision, additional papers have been published in the ALS/FTD field emphasizing the role of endolysosomal vesicular degradation pathways in ALS and ALS/FTD pathogenesis (e.g., Shao, *Science*, 2022; PMID: 36201573 and Todd, *Trends Neuroscience*, 2023; PMID: 37827960). The *Science* paper has been referenced and discussed within our revised manuscript.

Major

R1.2: the authors demonstrate that rapamycin reverts some phenotypic alterations of PFN1-mut microglial-like cells, and these results are presented as a demonstration of the role of autophagic process in some steps of lipid metabolism and phagocytosis alterations. It would be interesting to see the effects of rapamycin treatment on PFN1 mut mice models of ALS. Is rapamycin administration able to reduce disease progression in these animals? This information would significantly strengthen the data and would demonstrate the author's hypothesis of a role for microglia in PFN1-ALS.

Response R1.2: The mice reported herein represent a novel knock-in model harboring a heterozygous PFN1 C71G mutation (PFN1 C71G^{+/-}). Our intention for using this model was to assess phagocytic degradation of phagocytosed material (i.e., dead neurons) within an in vivo context to complement the outcomes from iMGs, which showed that mutations in PFN1 impair phagocytic processing. Data presented in **Figure 5** show that PFN1 C71G^{+/-} mice are deficient in phagocytic processing of dead neurons compared to their WT counterparts, indicating that the effects of mutant PFN1 on the endo-lysosomal pathway can be recapitulated in vivo. As suggested by Reviewer #2, comment #6 (**R2.6**), we revised **Figure 5b** with new confocal images from the dead-neuron injection study that more clearly demonstrate the accumulation of phagocytosed dead neuron material in mutant mice.

We agree with the Reviewer that additional data from in vivo models will strengthen the premise that disruption of vesicular degradation pathways by mutant PFN1 directly contributes to ALS pathogenesis. However, as is the case with other ALS and ALS/FTD knock-in mouse models or models with near-endogenous levels of mutant protein (i.e., models without transgene overexpression), the PFN1 C71G^{+/-} mice do not exhibit overt ALS-related phenotypes such as paralysis, motor neuron dysfunction or neuroinflammation (described in detail below). Examples of other models include mutant SOD1 (Dominov, *BioRxiv*, 2023; PMID: 37205335) and mutant C9ORF72 (Owens, *Neuron*, 2015; PMID: 26637797); we have included the Owens reference in our revised manuscript. Although knock-in models generally do not present with signs of overt neurodegeneration, they do present with pathophysiological phenotypes at the molecular and cellular level (e.g., impaired phagocytic processing) that observed during disease pathogenesis and that are unlikely to arise from overexpression artifacts.

As PFN1 C71G^{+/-} mice represent a novel mouse model generated at Jackson laboratories, we include new data generated with PFN1 C71G^{+/-} mice aged ~600 days; note that homozygous PFN1 C71G^{+/+} mice are embryonic lethal. As shown in **Supplemental Figure 12a,b**, an automatic capillary Western analysis demonstrated that PFN1 protein levels were significantly reduced in brain and spinal cord tissues from PFN1 C71G^{+/-} mice compared to their WT counterparts; these results are consistent with observations of reduced PFN1 expression in our mutant iMGs (Figure 2b in the current manuscript and previous literature, as referenced). In **Supplemental Figure 11c**, we show there are no statistically significant differences between PFN1 C71G^{+/-} and WT mice using the rotarod test, a standard test used to

assess neuromuscular coordination in ALS mouse models. There are also no statistically significant differences between genotypes in the grip strength test, another test used to assess ALS mice that measures motor function of the fore and hindlimb paws (**Supplemental Figure 11d**). We also examined brain and spinal cord tissues for PFN1 C71G^{+/-} and WT mice using antibodies reactive for Iba1 (microglia), CD68 (lysosomal compartments within microglia), and GFAP (astrocytes) as presented in **Supplemental Figure 13**. While there were subtle differences between PFN1 C71G^{+/-} and WT mice for some markers, none reached statistical significance. Therefore, we are unable to test whether rapamycin administration reduces disease progression in these animals. As a note to the Reviewer and inferred in the Discussion, we plan on continuing to probe PFN1 C71G^{+/-} mice for subtler phenotypes as part of future experimental plans. The results text has been revised as follows:

“Total PFN1 protein levels were reduced in PFN1 C71G^{+/-} mice compared to WT controls (Sup. Figure 11) as observed in iMGs (Figure 2). Consistent with other ALS and ALS/FTD mouse models lacking mutant protein overexpression (PMID: 26637797), overt phenotypes related to motor neuron dysfunction (Sup. Figure 11) and gliosis were not observed in PFN1 C71G^{+/-} mice aged to ~600 days (Sup. Figure 12). Nevertheless, PFN1 C71G^{+/-} mice provide an in vivo system to study the effects of PFN1 mutation on phagocytic processing.”

R1.3: In dead neuron injection experiments the authors used both male and female mice, but the results are shown together. Did they observe sex-related differences?

Response R1.3: *A priori*, we had no reason to suspect that sex would be a biological variable for these studies and therefore both male and female mice were included. Although there were not enough mice to perform a statistical analysis of sex differences in these studies, we have revised figures and graphs to include different symbols for male (open circles) versus female (closed circles). We have also included separate data source files showing male and female mice plotted separately. Specifically for the fraction of Cy5E-neuron (signal that is in acidic compartments) that overlays with the AF-neuron (total neuron signal), there are both male and female mice above and below the mean. Therefore, sex does not appear to be a confounder for these experiments showing enhanced dead neuron signals in acidic compartments within PFN1 C71G^{+/-} mice.

Minor

R1.4: More info about PFN and ALS should be given in the introduction. The connection is only weakly introduced at page 3 line 84, where the authors state that PFN1 mutations are causative for ALS, but they should better explain its incidence among ALS cases and if mutations of this gene give a 100% ALS disease, the severity of this form in comparison with fus, tdp43, sod1 and c9orf72-mut forms.

Response R1.4: We thank the Reviewer for this suggestion and have updated the text in the Introduction as follows:

“Herein, we used iMGs to investigate the role of profilin 1 (PFN1) in ALS, a fatal neurodegenerative disorder. PFN1 plays an essential role in modulating actin dynamics through interactions with actin, proteins enriched with poly-L-proline motifs, and phosphoinositide (PIP) lipids. Autosomal dominant mutations in PFN1 are highly penetrant and account for 1-2% of inheritable ALS. Further, cytoskeletal dysfunction is a major pathophysiological feature of both familial and sporadic ALS. ALS-PFN1 predominately presents with limb onset and a similar disease course as other forms of ALS, however the mechanism underlying PFN1-mediated ALS has not been elucidated.”

R1.5: The animal model used must be introduced. Which signs of the disease were present in the PFN1 C71G^{+/-} mice generated with CRISPR/Cas9 technology? When do they appear in the animal life?

Response R1.5: As described under **Response R1.2**, the PFN1 C71G^{+/-} knock-in mice do not exhibit overt ALS-related phenotypes such as paralysis, motor neuron dysfunction or neuroinflammation by the age (~600 days) the mice were assessed. That said, the phenotypes we uncover related to phagocytic processing in PFN1 C71G^{+/-} knock-in mice at 3 months of age can be caused by physiological expression of mutant PFN1, underscoring this as a physiological phenotype that can be recapitulated in vivo and without mutant protein overexpression.

Reviewer #2

Reviewer #2, comment #1 (R2.1): Funes et al. describe in their manuscript the effect profilin-1 (PFN1) mutation in human iPS cells derived microglia. Using CRISPR/Cas9 gene-editing they introduced ALS-linked C71G^{+/-}, M114T^{+/-} and M114T^{+/+} mutations in human induced pluripotent stem cells and then differentiating them into microglia like cells (iMG), they demonstrate deficit in phagocytosis and lipid metabolism. Further, they showed that enhanced binding of mutant PFN1 to the autophagy signaling molecule PI3P, as an underlying cause of defective phagocytosis. Although overall the work is important for understanding how PFN1 mutation affects microglial function, the study remains highly descriptive and correlative. It lacks, in my view, further mechanistic insights into how PFN1 mutation affects microglial phagocytosis.

Response R1.2: We thank the Reviewer for finding our work important and for their constructive comments. In the revised manuscript, we provide additional data and a new model **Figure 8h** to clarify our interpretation for how deficits in autophagic and endo-lysosomal processing manifest in mutant PFN1 iMGs. Through the revision process, we came to appreciate that our initial conclusions placed too much emphasis on autophagy *underlying* the phagocytosis deficits in mutant PFN1 iMGs. As we illustrate in **Figure 8h** and discuss throughout the manuscript, autophagy and phagocytosis are distinct yet interconnected pathways with shared degradative machinery. Our data support a model whereby mutations in PFN1 impede vesicular degradation in *both* autophagy and phagocytosis pathways, and that the effects of rapamycin likely ameliorate deficits in both pathways. We acknowledge here and in the manuscript that we do not yet know precisely how mutant PFN1 impedes vesicular degradation, but it is likely through a gain-of-toxic mutant PFN1 function (rather than a loss of normal PFN1 function). Our discovery that mutant PFN1 exhibits enhanced binding to PI3P, a critical signaling lipid involved in *both* autophagy and phagocytosis raises the intriguing possibility that mutant PFN1 interferes with vesicular degradation through aberrant interactions with PI3P and potentially other PIPs, which we intend to explore further in our future work.

Major concern

R2.2: In figure 1. authors assume LPS induced production of cytokines in PFN1 mutation carrying iMG and control is similar using C71G^{+/-} only. However, the data regarding M114T mutation carrying iMG is missing. Could M114T mutation have similar response?

Response 2.2: We include new results of LPS-induced cytokine production in M114T^{+/-} and M114T^{+/+} iMGs in **Supplemental Figure 3**. The results with the M114T lines largely agree with C71G^{+/-} iMGs shown in the main text (Figure 1e) and confirm that M114T^{+/-} and M114T^{+/+} iMGs are reactive to LPS. There is a modest yet significant increase in IL-6 secretion only for M114T^{+/-} iMGs after 6h of LPS exposure, and a moderate yet significant decrease in IL-10 secretion for M114T^{+/+} iMGs at baseline. Collectively, our interpretation is that there are no major effects of PFN1 mutation on secretion of this subset of cytokines. The text in the results section is revised as follows:

“The cytokine levels were generally similar between mutant and WT PFN1 cells, except IL-6 was moderately elevated in M114T lines upon LPS stimulation and IL-10 reduced in M114T^{+/+} at baseline (Figure 1e, Sup Fig. 3 and Table S1). Collectively, these data show that iMGs from both genotypes are responsive to immune stimulation.”

R2.3: In figure 2, authors first showed that C71G^{+/-}, M114T^{+/-}, M114T^{+/+} mutation cause PFN1 protein decrease by 60, 30 and 80% respectively. However, lipid accumulation increase observed in C71G^{+/-}, M114T^{+/+}, was not observed in M114T^{+/-}. Is there any correlation between amount of PFN1 decrease and accumulation of lipid droplet in iMG? Also, from the data presented in current manuscript is insufficient to indicate whether the autophagosome dysfunction is due to decreased PFN1 protein or due to abnormal protein binding. So, it advised that, authors should perform knockdown of PFN1 in iMG or HMC3 cell line to show its effect on to lipid accumulation.

Response 2.3: We thank the Reviewer for this suggestion and agree that experiments addressing loss versus gain of PFN1 function are of interest. We present new data with knocked-down PFN1 expression in HMC3 cells in **Supplemental Figure 5**, which shows that reduced PFN1 expression does not lead to enhanced lipid droplet formation. During the revision process, we noticed less fluorescence background when probing for lipid droplets using an anti-perilipin-2 antibody (where perilipin 2 is a prominent lipid

droplet associated protein) compared to the BODIPY dye and opted to carry out the HMC3 study with an anti-perilipin antibody.

The observations in HMC3 cells also support the premise that enhanced lipid droplet formation in mutant PFN1 iMGs is not due to a loss-of-function, but rather a gain-of-function due to the mutant protein itself. We reported (i.e., Boopathy, *PNAS*, 2015) that the C71G mutation most severely destabilizes the PFN1 structure, as evidenced by the apparent free energy of folding (ΔG°) of 1.89 kcal/mol for this variant compared to 7.0 kcal/mol for PFN1 WT. PFN1 M114T is in between C71G and WT PFN1, at 3.51 kcal/mol. This same correlation is observed for lipid droplet content (Figure 2h-j): M114T het < C71G het < M114T Hom. Therefore, the difference between M114T het and M114T hom is likely a consequence of *more mutant* PFN1 M114T, rather than decreased levels of total PFN1, in the Hom line. These points are also relevant to **R2.3** and **R2.9** below and are addressed in the revised results text as follows:

“...all mutant PFN1 iMGs expressed lower PFN1 levels compared to control cells by Western blot analysis, with ~60% reduction in PFN1 C71G^{+/-} iMGs (Figure 2b,c) and ~30% reduction in PFN1 M114T^{+/-} iMGs (Figure 2b,d). Strikingly, PFN1 levels were reduced by 80% in M114T^{+/+} iMGs compared to WT iMGs (Figure 2b,d). These results are consistent with the destabilizing effect of ALS-linked mutations on PFN1 structure, where the C71G variant is more misfolded than the M114T variant and thus is more robustly degraded by the proteasome”

and

“Next, PFN1 knockdown studies were pursued to determine whether lipid droplet accumulation is a consequence of PFN1 loss-of-function. As lentiviral transduction was toxic to iMGs, these studies were carried out in the human microglia immortalized HMC3 line using an antibody against perilipin 2, a prominent lipid droplet-associated protein. Lipid droplet content was not elevated or significantly different upon PFN1 knockdown compared to control cells (Sup Fig. 5), implicating a gain-of-toxic function for mutant PFN1 with respect to lipid dysmetabolism in iMGs. Taken together, the trend in M114T^{+/-} < C71G^{+/-} < M114T^{+/+} iMGs for lipid droplet content likely reflects that C71G^{+/-} is more misfolded than M114T^{+/-} and that the relative levels of misfolded protein are highest in M114T^{+/+} iMGs.”

R2.4: Authors have shown that C71G^{+/-}, M114T^{+/-}, M114T^{+/+} mutation in PFN1 increase TBC1D15 in fig. 3. Authors also show that both wild type mutant and PFN1 show interaction with TBC1D15 by immunohistochemistry. However, PFN1 knockdown in human microglial cell line HMC3 leads to decrease in TBC1D15. Although this upregulation of TBC1D15 in PFN1 mutant iMG led authors to further investigate phagocytotic pathway, the explanation and data they have provided to this discrepancy remains unsatisfactory. First of all, the interaction between TBC1D15 and PFN1 remains unknown. Authors can determine this interaction as they have done in figure 8. Next the authors can determine how PFN1 mutation affect its interaction as well as degradation of TBC1D15 to clarify its role in the processing of phagocytotic products. Although authors have stated that the upregulation of TBC1D15 as a compensatory mechanism the defect in the processing of phagocytotic products, it may well be the cause of this dysfunction. This issue needs to be addressed.

Response R2.4: In the original manuscript, we did not intend to imply that PFN1 and TBC1D15 directly interact. Both Reviewers #2 and #3 requested clarification on the relationship between PFN1 and TBC1D15, as well as the relevance of TBC1D15 upregulation in mutant PFN1 iMGs.

In revised **Figure 3**, we show new immunofluorescence analyses that TBC1D15 and the outer mitochondrial membrane marker TOMM20 co-localize. Further, TBC1D15 and TOMM20 co-localization is significantly enhanced in mutant C71G^{+/-} iMGs compared to control iMGs. Previous literature from Dr. Richard Youle's lab (PMID: 24569479) demonstrated that TBC1D15 is an important factor in mitophagy, a form of selective autophagy that removes old and/or damaged mitochondria. Specifically, TBC1D15 binds to and regulates proper autophagosome formation around damaged mitochondria. That TBC1D15 levels positively correlate with mitophagy suggests that upregulation of TBC1D15 favors autophagy. In addition to TBC1D15, we also show new data validating the upregulation of a second autophagy/mitophagy factor, **RASD2**, that emerged from our RNASeq analysis (this gene is now emboldened in the RNASeq volcano plot in revised **Figure 3a**). In **Supplemental Figure 6g,h**, we show that RASD2 is upregulated at the protein level. In **Supplemental Figure 8d,e**, we show new data that total levels of mitochondria assessed with anti-VDAC (voltage-dependent anion-selective channel protein) were not significantly different

between PFN1 C71G^{+/-} and control iMGs. This entire results section has been revised and we conclude with:

“Therefore, there appears to be an increase in signals marking mitochondria for autophagosomal degradation in PFN1 C71G^{+/-} iMGs, but without an overall change in mitochondrial content.”

In the revised discussion section, we interpret upregulation of TBC1D15 and RASD2 at the mRNA and protein levels as an attempt by mutant PFN1 iMGs to upregulate autophagic processes, and not as the cause of microglial dysfunction.

Regarding the Reviewer’s comment “Authors can determine [PFN1 and TBC1D15] interaction as they have done in figure 8”, the predicted molecular weight of TBC1D15 protein is 79 kDa, which when bound to PFN1 (~15kDa) exceeds the size limitation for the NMR titration experiments shown in Figure 8. While PFN1 and TBC1D15 also exhibit enhanced co-localization in mutant iMGs, PFN1 and TOMM20 do not. Our interpretation is that

“It is therefore unlikely that PFN1 directly increases TBC1D15/TOMM20 co-localization in mutant iMGs.”

We also note that in the revised manuscript, we used a different immunofluorescence protocol that contained less detergent to better preserve mitochondrial-associated proteins for assessment of TOMM20 and TBC1D15. Using this protocol, the colocalization signals between TBC1D15 and PFN1 in mutant iMGs were still statistically significant (**Supplemental Figure 8b**), however the appearance of the staining is more diffuse and less punctate. We believe this is because the original immunofluorescence protocol removed more of the diffuse TBC1D15 signals, leaving behind punctate structures containing PFN1 and TBC1D15 that were more obvious than with the current staining protocol. For simplicity and to focus on the mechanistic insight provided by TBC1D15 localization to mitochondria, we only include data from the immunofluorescence protocol tailored to TOMM20 in the revised manuscript.

R2.5: Authors have assumed that all PFN1 mutation affects iMG in the same way even though their own data contradicts this assumption as shown in figure 2 i. They have generalized that since C71G^{+/-} mutant iMG do not have defect in synaptosome uptake, it must be true for M114T^{+/-}, M114T^{+/+} mutant iMG. However, the it may not true. So, it is advisable that they include the data for M114T^{+/-}, M114T^{+/+} mutant iMG in Figure 4. Also, from the graph presented in Figure 4k, it is hard to appreciate difference as they have shown.

Response R2.5: We maintain that for every figure with data for both PFN1 C71G and M114T variants, the mutation-induced phenotypes are consistent with each other. As discussed under **Response 2.3**, mutation-induced phenotypes may follow the following trend due to severity of the mutation and amount of mutant protein that is expressed: M114T het < C71G het < M114T Hom. As we noted in the manuscript, *“the homozygous M114T^{+/+} variant was created as an experimental line to investigate mutant-gene dosage on potential phenotypes”*, since humans with ALS harbor heterozygous mutations in PFN1.

Below, we list the experiments with both PFN1 C71G and M114T variants, and comment on their consistency. **Responses that are specific to R2.5 are in blue font:**

Supplemental Figure 1: PFN1 C71G and M114T iPSC lines are consistent in terms of pluripotency marker expression. There are no mutant-specific phenotypes.

Figure 1 and revised Supplemental Figure 2: PFN1 C71G and M114T iMGs are consistent in terms of microglia and macrophage marker expression. There are no mutant-specific phenotypes.

Figure 1 and new Supplemental Figure 3: PFN1 C71G and M114T iMGs are consistent in terms of cytokine secretion in response to LPS. M114T iMGs exhibit some moderate differences in cytokine secretion as addressed in **Response 2.2.**

Figure 2i: Lipid droplets accumulate in mutant PFN1 iMGs. Lipid droplet phenotype correlates with genotype as follows: M114T het < C71G het < M114T Hom, which is fully addressed in **Response 2.3.**

Figure 3: TBC1D15 upregulation is detected in both PFN1 C71G and M114T iMGs, at both the mRNA and protein levels.

Figure 4 and new Supplemental Figure 10: Both PFN1 C71G and M114T iMGs exhibit inefficient phagocytic degradation.

In response to the Reviewer, we revised Figure 4k (Figure 4l in the current version), where data from the mutant line is now normalized to the respective WT control line from the same experiment; the effect of the PFN1 mutation is more obvious in the revised graph.

In response to the Reviewer, we measured synaptosome uptake in M114T^{-/-} and M114T^{+/+} iMGs. Because our original synaptosome uptake assessment was labor intensive, we developed a flow-based assay to measure synaptosome uptake as described in the revised methods “Flow cytometry synaptosome uptake assay and analysis”. First, we demonstrated the same outcome with the flow-based assay (new **Figure 4i and Supplemental Figure 10**); as with the original analysis (**Figure 4h**) with C71G^{+/-}; both methods show the C71G mutation does not affect synaptosome uptake. Next, we assessed both M114T^{-/-} and M114T^{+/+} iMGs with the flow-based assay and found that synaptosome uptake was not affected by either genotype as shown in **Supplemental Figure 10**.

Figure 7: Phagocytic degradation of material is restored to WT levels with rapamycin treatment in both PFN1 C71G and M114T iMGs. Based on Figure 4, we knew that C71G^{+/-} and M114T^{-/-} lines exhibit similar degrees of phagocytic dysfunction and therefore opted to combine the heterozygous genotypes (denoted as “ALS-PFN1”; further discussed in **Response R2.12**) while keeping the experimental, homozygous line separate.

Figure 8: New data are included to show that PFN1 C71G binds to PI3P as shown for M114T. As discussed in **Response R2.3** and below in **Response R2.9**, the C71G mutation induces severe misfolding in PFN1 that precludes quantification of the binding constant between PFN1 C71G and PI3P by NMR.

R2.6: Using PFN1 C71G^{+/-} mice authors have shown that higher number of dead neurons in microglial acidic compartments in figure 5. Further they showed that microglia in PFN1 C71G^{+/-} mice that showed increased dead neuron in phagosome and higher expression of Iba1 (figure 5b-e). Is the number of microglia and Iba1 intensity between PFN1 C71G^{+/-} and control mice similar at basal condition?

Response R2.6: This is addressed and fully discussed in **Response R1.2**; there is no difference in Iba1 intensity between PFN1 C71G^{+/-} and control mice at basal condition.

R2.6: Also, it is hard to appreciate that higher number of dead neurons in microglial acidic compartments of PFN1 C71G^{+/-} microglia from the image presented in fig 5 b, it would be nice if they could present more convincing image to further support their data.

Response R2.6: The original images were acquired on a wide-field fluorescence microscope. In response to the Reviewer’s comment, we acquired new images from the same slides on a confocal microscope and present these in revised **Figure 5e**. We believe that the confocal improved the quality and clarity of these images.

R2.6: Also, what cause this decrease in microglia in ring 3 region in PFN1 C71G^{+/-} mice?

Response R2.6: We added text to clarify our speculation that migration of microglia from the region within Ring 3 towards the site of dead neurons causes this change, which may also account for increased microglia in Rings 1 and 2 within the mutant mice.

R2.6: Also do microglia in these mice show accumulation of lipid? increase in TBC1D15? defect in phagosome processing?

Response R2.6:

Our interpretation of enhanced signals located within acidic compartments (**Figure 5**) is that there are defects in phagosome processing. We agree with the Reviewer that it would be of interest to probe mice with injected neurons for other phenotypic changes, including lipid accumulation and enhanced TBC1D15 expression. All the brain tissue from the initial study was used for the analysis in **Figure 5**; there was no tissue remaining. To address the Reviewer’s inquiries, we generated a new cohort of PFN1 C71G^{+/-} and WT littermate mice and repeated the dead neuron injection study. For this second study, we used a new stereotactic injection device that unfortunately failed for half the mice; when we processed the tissue, we discovered that for many of these mice the dead neurons did not inject into the brain tissue but remained at the surface of the brain. This was a significant undertaking in time and resources, and a regret that we were unable to address these questions within a reasonable time frame.

R2.7: Additionally, is Iba1 expression in iMG increased when they are presented with dead neurons or synaptosome?

Response 2.7: Iba1 levels do change as a function of phagocytosis of synaptosomes in iMGs but this change is not significantly different between genotypes as presented in new **Supplemental Figure 10d**.

R2.7: Authors have shown that rapamycin attenuate autophagic as well as phagocytotic dysfunction in PFN1 mutant iMG. Do addition of rapamycin normalize EEA1 expression as well as LC3II/LC3I ratio?

Response R2.7: In our original manuscript we showed a difference in LC3I/LC3II ratio between mutant and WT iMGs by Western blot analysis. During the revision process, we did not observe a difference LC3II/LC3I ratio. The reason for this discrepancy is unclear, although we note that measuring LC3I/LC3II levels by Western blot was extremely difficult in iMGs in our hands during the initial phase of this project, much more so than any other Western blot analysis we perform in the lab. For LC3I/LC3II, we tried two different transfer methods (dry versus wet), different membranes, buffers and anti-LC3 antibodies. During the revision phase of the project, we attempted LC3I/LC3II on n=5 independent iMG differentiations using our most reproducible protocol; the outcomes indicate no difference in LC3I/LC3II (**Figure 7a-c**) but still show a significant increase in p62 (**Figure 7d,e**). In the revised manuscript, we pursued new immunofluorescence analyses of LC3 and p62 as an orthogonal approach to the Western blot. As shown in new **Figure 7f-j**, the outcomes of the immunofluorescence agree with the Western blot analyses. Thus, while LC3I/LC3II does not change between genotypes, our cumulative data still point toward deficits in autophagic processing in mutant PFN1 iMGs. The Results section pertaining to Figure 7 has been revised accordingly.

During the revision phase, we did not have the additional bandwidth to repeat the EEA1 analysis (**Supplemental figure 14a**) with Rapamycin. We plan to further define the effects of rapamycin in mutant iMGs through omics analyses in our future experiments.

R2.8: Also, in addition to graph, authors should include the immunocytochemistry image in the main or supplementary figure to show that rapamycin treatment affect phagocytotic index.

Response R2.8: We have included an image from the live imaging analysis that illustrates differences +/- Rap in new **Supplemental Figure 14b**.

R2.8: The information regarding how is AUC calculated in Fig 7 i, is missing. It should be included in the method section.

Response R2.8: The methods section now describes how we calculated AUC: "*The area under the curve was determined using GraphPad Prism v9.3.1*".

R2.9: As mentioned above, authors assumed that all PFN1 mutation affects iMG in the same way, for protein-protein interaction with PI3P, they have used M114T mutation and generalized this data to all mutation affecting PFN1. As mentioned above each mutation may affect PFN1 function in microglia in different way, so the effect of each mutation should be individually assessed. Why do authors use only M114T mutation for protein-protein interaction instead of C71G? Authors should include PFN1 C71G data to show that similar interaction occurs PI3P.

Response R2.9: As discussed in **Response R2.3** and **Response R2.5**, isolated C71G is unstable and difficult to work with in vitro. In our previous two manuscripts (Boopathy, *PNAS*, 2015 and Schmidt, *PNAS*, 2021) we could not generate sufficient PFN1 C71G for structural analyses such as the NMR shown in Figure 8. Less material is required for differential scanning fluorimetry (DSF) in Figure 8, and therefore we now include new data in **Supplemental Figure 15a** showing that addition of PI3P induces the same effects on C71G as the other variant M114T. Because PFN1 C71G is so destabilized, the 200 μ M PI3P used for the DSF caused complete denaturation of the protein. Therefore, even if we attempted the NMR studies with PFN1 C71G, the protein is expected to precipitate out during the titration of PI3P. The text has been revised in the results section as follows:

*"In the absence of ligand, the T_m of PFN1 M114T and C71G is, respectively, $\sim 14^\circ\text{C}$ (**Figure 8a**) and $\sim 20^\circ\text{C}$ (**Sup. Fig 15**), below that of PFN1 WT, consistent with a destabilizing effect of these mutations on PFN1 with C71G being more severe. Increasing concentrations of PI3P caused a reduction in the T_m of all PFN1 proteins (**Figure 8a-c**). Upon addition of 200mM PI3P, PFN1 M114T exhibited a significantly larger decrease in T_m (ΔT_m) than PFN1 WT (**Figure 8a-c**) and C71G unfolded to the extent that a proper binding curve was not observed (**Sup. Fig 15**). Due to the severe instability of PFN1 C71G in vitro, subsequent experiments were pursued with PFN1 M114T."*

R2.10: Authors should perform co-culture to show that PFN1 mutation in microglia affect neurons under basal and stressed condition to show the to clarify how microglia dysfunction affect neurons in non-cell autonomous way.

Response R2.10: We agree with the Reviewer that these are logical and important next steps. Certainly, it will be interesting to investigate whether mutant PFN1 iMGs affect human neurons. However, the focus of the current manuscript was to report on how disease-linked genes such as ALS-PFN1 mutations affect the intrinsic properties of microglia. Therefore, we view these co-culture studies as being outside the scope of the current manuscript.

Minor points

R2.11: There are two GAPDH band in figure 7 c. Please remove one GAPDH band.

Response R2.11: Thank you for pointing this out; this has been addressed.

R2.12: Also which mutation do authors mean by ALS-PFN1 in 7 h. Please change it to C71G^{+/-} for consistency so that is easy for readers to understand.

Response R2.12: We apologize for this confusion. As noted in **Response R2.5**, by the time we conducted experiments for Figure 7, we knew that the heterozygous lines (C71G and M114T) behaved similarly in the context of the phagocytosis assay and therefore we combined data for these lines. This is defined in the figure legend, and we now use different symbols for the M114T and C71G lines to enhance transparency for the Reader. As one can see from the graphs, the data from both genotypes are similar and overlap around the mean.

R2.13: The authors have stated that “Area under the curve (AUC) determined from 7 i” in figure 7 i. This mistyping should be rectified.

Response R2.13: Thank you for pointing this out; this has been addressed.

Reviewer #3

In this paper by Funes et al., the consequences of PFN1 mutations in iPSC-derived microglial-like cells are studied. As PFN1 is expressed at higher levels in microglia than neurons, and microglial activation is a hallmark of ALS pathology the study is relevant to the field. The engineered mutant lines (with two different mutations) differentiate with normal efficiency into microglial cells and displayed similar activation in response to immune stimulation. However, they display some changes in proteome/transcriptome and a problem of clearance of vesicles.

Reviewer #3, comment #1 (R3.1): The experiments are clearly presented, a large quantity of data is presented. Several observations are made including an accumulation of lipid droplets, increased expression of TBC1D15, perinuclear accumulation of perinuclear LAMP1-positive vesicles, reduced autophagic degradation of vesicles and increased PIP3 binding with reduced PFN1 stability.

The relation between these different phenotypes is not always clear and makes the paper heavy to digest.

Response R3.1: We thank the Reviewer for their positive comments and appreciate that we needed to clarify the relation between the phenotypes presented in the paper. To this end, we performed additional experiments to gain greater insight into the mechanisms of how mutant PFN1 affects vesicular transport within microglia and have created a **new Figure 8h** to illustrate our proposed model. This model is discussed in **Response R1.2**.

Specific comments:

R3.2: Figure 1: the microglial markers and the responsiveness of the iMGs are only shown for the C71G^{+/-} mutants, but not for the M114T^{+/-} and ^{+/+} lines. It would be of interest to add this as well as it provides extra gene-dose information.

Response R3.2: In the original manuscript, we presented an RNASeq dataset for PFN1 WT, C71G^{+/-} and M114T^{+/-} lines. From this RNASeq dataset, we extracted the normalized read counts of genes associated with different microglia developmental stages expressed in PFN1 WT, C71G^{+/-} and M114T^{+/-} iMGs as well as the normalized read counts of the microglia-enriched genes in PFN1 WT, C71G^{+/-} and M114T^{+/-} iMGs. This data is now presented in **Supplemental Figure 2b and c**, respectively, and complement the data presented in Figure 1. None of the genes exhibited significant differential expression between genotypes (genes with significant differential expression were presented in the volcano plot in Figure 3a), and the

expression of these genes follow the pattern that is expected for microglia-like cells for all genotypes assessed.

We also include new cytokine ELISA results for PFN1 M114T^{-/-} and M114T^{+/+} iMGs upon LPS stimulation, which is discussed in detail under **Response 2.2**.

R3.3: A proteome and transcriptome analysis have been performed. The results have been analyzed and presented separately in the paper. A combined analysis would be of interest. Where some of the changes at protein level visible at transcript level as well? and vice versa?

Response 3.3: In our view, with only 13 significantly differentially expressed genes from the RNASeq analysis, performing a combined analysis with the proteomics did not seem practical. We note that while there were no differentially expressed factors in common to both analyses, factors from both analyses were enriched in common pathways. Notably, *TBCD15* and *RASD2* from the RNASeq analysis (**Figure 3a**) are both involved in vesicular degradation, including autophagy and mitophagy (selective autophagy of mitochondria). Further, “phagosome” (component of vesicular degradation) as well as “mitochondrial pathway of apoptosis” are highlighted in the pathway analyses from the TMT proteomics (**Figure 2** and **Supplemental Figure 4**). In the revised manuscript, we show new data for the validation of *RASD2*, which is discussed in detail in **Response R2.4**. We also note that RNASeq and proteomics analyses often do not reveal the same factors, which can be due to technical reasons (i.e., some proteins do not ionize well for mass spectrometry) and/or biological reasons (some genes change at the mRNA level but not at the protein level, and vice versa).

R3.4: Figure 3: based on the staining experiments, the authors conclude that there is colocalization of *TBC1D15* and *PFN1*. In the images only a partial colocalization is visible. What could be the importance of this finding? How is it linked to what is presented in later figures in the paper?

Response R3.4: In the revised manuscript, we present new data showing that upregulation of *TBC1D15* correlates with enhanced co-localization of *TBC1D15* with mitochondria in mutant *PFN1* iMGs (revised **Figure 3**). While there is also enhanced colocalization between *PFN1* and *TBC1D15* in mutant iMGs, our interpretation is that mutant *PFN1* expression indirectly leads to enhanced *TBC1D15* association with mitochondria, presumably in the context of mitophagy. Therefore, we have placed less emphasis on the co-localization between *TBC1D15* and *PFN1* in the current manuscript and focus more on *TBC1D15* and mitochondria. We have discussed this in detail in **Response R2.4**.

R3.5: Figure 4: Bafilomycin abolishes the pHrodo signal as expected. Was there an effect on the engulfment index and on the volume occupied by pHrodo-synaptosome signal?

Response R3.5: As discussed in under **Response R2.5**, our original synaptosome uptake assessment was labor intensive and therefore we developed a flow-based assay to measure synaptosome uptake as described in the revised methods “Flow cytometry synaptosome uptake assay and analysis”. First, we demonstrated the same outcome with the flow-based assay (new **Figure 4i** and **Supplemental Figure 10**) as with the original analysis (**Figure 4h**) with *C71G^{-/-}*; both methods show the *C71G* mutation does not affect synaptosome uptake. Next, we assessed both *M114T^{-/-}* and *M114T^{+/+}* iMGs with the flow-based assay and found that synaptosome uptake was not affected by either genotype as shown in **Supplemental Figure 10**. Per the Reviewers inquiry, we also included a BafA condition in these experiments and revised the results section as follows:

“This assay also showed that BafA attenuates but does not prevent synaptosome uptake (Figure 4i and Sup. Fig. 10). Therefore, lack of pHrodo fluorescence in the presence of BafA (Figure d,e) can be attributed to inhibition of lysosome acidification and not due to inhibition of synaptosome uptake.”

R3.6: Figure 7: was there still an increase in LC3-II and p62 upon bafilomycin treatment?

Response R3.6: As discussed in detail in Response 2.7, differences in LC3I/II were not observed during the revision process. Increased levels of p62 were observed in mutant iMGs under baseline conditions; this data is now presented in Figure 7d,e and the results was revised as follows:

*“Bafilomycin A treatment led to an overall increase in p62 levels, but without a significant difference between genotypes, indicative of reduced autophagic processing in untreated *PFN1 C71G^{-/-}* iMGs (Figure 7d-e).”*

REVIEWERS' COMMENTS

Reviewer #1 (Remarks to the Author):

The authors addressed all my concerns in an acceptable manner inserting additional information on the animal model. The considerations on the use of males and females mice reveal one weak point of the study that needs to be addressed.

Reviewer #2 (Remarks to the Author):

No further comment.

Reviewer #3 (Remarks to the Author):

My comments have been sufficiently addressed.